METHODS AND RESOURCES

# DRfold2 is a deep learning-based tool that enables efficient and accurate RNA structure prediction

Yang Li[1], Chenjie Feng[2], Xi Zhang[3], Sho Tsukiyama[4,5], Duanyu Feng[1], Yang Zhang[1,3,4,6]*

1 Cancer Science Institute of Singapore, National University of Singapore, Singapore, Singapore, 2 School of Science, Ningxia Medical University, Yinchuan, China, 3 Center for AI and Computational Biology, Suzhou Institute of Systems Medicine, Chinese Academy of Medical Sciences & Peking Union Medical College, Suzhou, China, 4 Department of Computer Science, School of Computing, National University of Singapore, Singapore, Singapore, 5 School of Life Science and Technology, Institute of Science Tokyo, Tokyo, Japan, 6 Department of Biochemistry, Yong Loo Lin School of Medicine, National University of Singapore, Singapore, Singapore

* zhang@nus.edu.sg

## Abstract

RNA structures are essential for understanding their biological functions and developing RNA-targeted therapeutics. However, accurate RNA structure prediction from sequence remains a crucial challenge. We introduce DRfold2, a deep learning framework that integrates a novel pre-trained RNA Composite Language Model (RCLM) with a denoising structure module for end-to-end RNA structure prediction. Based solely on single sequence, DRfold2 achieves superior performance in both global topology and secondary structure predictions over other state-of-the-art approaches across multiple benchmark tests from diverse species. Detailed analyses reveal that the improvements primarily stem from the RCLM's ability to capture co-evolutionary pattern and the effective denoising process, with a more than 100% increase in contact prediction precision compared to existing methods. Furthermore, DRfold2 demonstrates high complementarity with AlphaFold3, achieving statistically significant accuracy gains when integrated into our optimization framework. By uniquely combining composite language modeling, denoising-based end-to-end learning, and deep learning-guided post-optimization, DRfold2 establishes a distinct direction for advancing ab initio RNA structure prediction.

## Introduction

Understanding the structure and function of RNA molecules has been a central focus of molecular biology and the pharmaceutical industry. RNAs, particularly non-coding RNAs, fold into specific structures that can be functional in various cellular processes, including gene regulation (e.g., transcription and translation), catalysis, cellular signaling, stress responses, and genome defense [1]. Due to their functional

**Data availability statement:** The datasets collected and used in this work are available at https://zhanggroup.org/DRfold2/data.zip and https://doi.org/10.5281/zenodo.18401691. All underlying data are available via the Supporting information file (S1 Data). We show structures of 7QR3, 8FZA, and 8DP3 obtained by four-digit accession codes in the PDB repository (https://www.rcsb.org/). All experimental RNA structures tested in this study are publicly available in the Protein Data Bank with the following accession codes: 7ELP, 7QR3, 7QR4, 7V9E, 7YR6, 7YR7, 8BU8, 8DP3, 8FZA, 8GXC, 8HB8, 8HZD, 8HZL, 8ITS, 8JHP, 8QO2, 8QO3, 8S95, 8SYK, 8T2P, 8TVZ, 8U5Z, 8UO6, 8V1H, 8VCI, 8VT5, 8XTR, 8XZL, 8Z8Q, 8Z8U, 9BCI, 9BUN, 9BZC, 9DE6, 9DXL, 9EOW, 9G7C, 9IS7, 9KPO. CASP16 results are available through the CASP prediction center (https://predictioncenter.org/). The RNA sequence data for RCLM training was download from RNAcentral (https://rnacentral.org/). The online server and standalone package of DRfold2 are freely available at https://zhanggroup.org/DRfold2/ and https://doi.org/10.5281/zenodo.18401691.

**Funding:** This work was supported in part by the Ministry of Education, Singapore (T1251RES2309 and T2EP20125-0039 to YZ), the Agency for Science, Technology and Research (A*STAR), Singapore (IAF-PP H25J6a0034 to YZ), and Natural Science Foundation of Ningxia Province (2023AAC05036 to CF). The funders had no role in study design, data collection and analysis, decision to publish, or preparation of the manuscript.

**Competing interests:** The authors have declared that no competing interests exist.

**Abbreviations:** AUC, area under the curve; DI, Deformation Index; DRSM, Denoising RNA Structure Module; IPA, Invariant Point Attention; LM, language model; MCQ, Mean of Circular Quantities; MSA, multiple sequence alignment; PCC, Pearson Correlation Coefficient; RCLM, RNA Composite Language Model; RMS, root mean square; RMSD, Root Mean Square Deviation.

significance, RNAs have emerged as promising targets for small-molecule drug development, especially for human diseases that lack traditional protein targets [2,3]. With the exponential growth of RNA sequence data from high-throughput sequencing and the widening gap between known sequences and experimentally resolved RNA structures, determination of atomistic structures of RNAs from the primary sequences has become increasingly urgent.

Researchers have developed multiple approaches to study RNA structures, ranging from chemical probing [4] and proximity ligation methods [5] for detecting base pairs or other interactions, to structural biology techniques such as X-ray crystallography [6], NMR spectroscopy [7], and cryo-EM [8]. Despite higher resolution, experimental RNA 3D structure determination is often prohibitively expensive and sometimes infeasible. Consequently, computational approaches have faced growing demands for high-quality 3D structure modeling directly from sequences.

Computer-based RNA structure prediction has evolved considerably over the past decades. Traditional template- and fragment-based approaches [9–12] construct structure models using known templates or fragments but are constrained by the limited availability of experimentally solved structures. Ab initio methods [13–15] attempt to predict structures from sequence alone; however, traditional physical or statistical force fields often lack the accuracy and specificity needed to fold RNAs with complex topologies. Recent advances in deep learning have introduced several innovative approaches. For example, DeepFoldRNA [16] and trRosettaRNA [17] utilize transformer-based models to predict local geometries and reconstruct coordinates from the predicted restraints. On the other hand, end-to-end models like RhoFold [18] and RoseTTAFoldNA [19] directly predict 3D coordinates, inspired by AlphaFold2 [20] in protein structure predictions. Our previous approach, DRfold [21], combines both end-to-end and geometry-based predictions through optimizing a hybrid potential function with a differentiable fold program [22].

Despite the progress, RNA structure prediction accuracy has not yet reached a so-called "AlphaFold moment" [23], partly due to limited solved structures compared to proteins. Language models have shown promise in protein structure prediction by capturing key features like co-evolution patterns and improving sequence representation from massive unsupervised sequence data [24]. Their ability to predict structures from single sequences is particularly valuable for RNAs, while multiple sequence alignment (MSA) construction [25] is more resource-intensive and the usefulness often varies depending on the depth of MSAs.

In this study, we introduce DRfold2, a new deep learning-based framework for high-quality RNA structure prediction. The core novelty of DRfold2 consists of a pre-trained RNA Composite Language Model (RCLM), which improves likelihood approximation and captures co-evolutionary signals from unsupervised RNA sequences more effectively than the previously used embeddings learned from structure prediction pipeline [21]. Next, the sequential and pairwise representations from RCLM are processed by an RNA Transformer-based pipeline, powered with a denoising module for end-to-end structure and geometry prediction. A post-processing protocol is then implemented for final RNA model selection and optimization, built on the end-to-end

and geometry potentials. The large-scale benchmark results demonstrate that co-evolution features learned by RCLM from unsupervised data, when integrated with denoising structure learning, significantly enhance the accuracy of ab initio RNA structure prediction.

## Results

The overall DRfold2 pipeline for RNA structure prediction is illustrated in Fig 1A. Starting from an RNA sequence, DRfold2 first employs a pretrained RCLM to embed the query into sequence and pair representations. Here, RCLM achieves superior sequence pattern recognition through training on large-scale unsupervised sequence data using a composite likelihood [26] maximization approach (Fig 1B). These sequence and pair representations are processed by a set of RNA Transformer Blocks (Fig 1C), which produce necessary representations for structure folding. DRfold2 then generates RNA conformations through a Denoising RNA Structure Module (DRSM) in an end-to-end fashion (Fig 1D). The final RNA models are obtained through a post-processing CSOR protocol that is designed to select and refine conformation decoys generated from a pool of checkpoints (Fig 1E).

To objectively benchmark the performance of DRfold2, we constructed an independent test dataset containing 41 RNA structures with sequence length < 400 nts collected from (1) recent RNA-Puzzles targets [27], (2) CASP15 RNA targets [28,29], and (3) recently released RNA-only structures after Year 2024 (before March 25th 2025) from Protein Data Bank

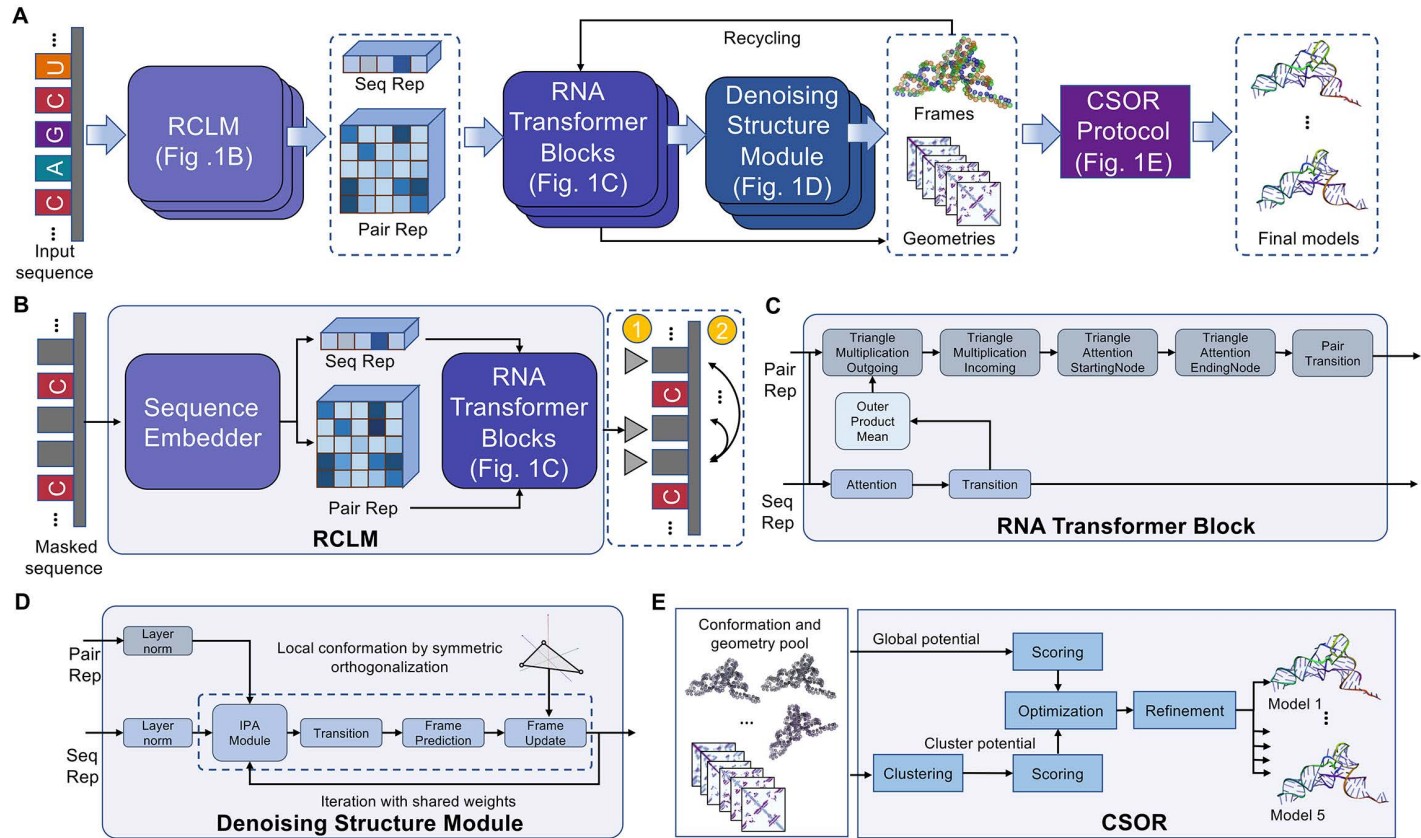

**Fig 1. The pipeline of DRfold2 for end-to-end RNA structure prediction. (A)** Overview of DRfold2 pipeline. **(B)** Details of training RCLM with masked negative composite log likelihood loss function. **(C)** Details of RNA Transformer Block. **(D)** Details of Denoising RNA Structure Module. **(E)** Detailed pipeline of CSOR Protocol as post-process to select and refine final RNA models.

(PDB) [30]. Notably, large synthetic RNAs from CASP15 were excluded due to their deviation from naturally occurring RNAs, which serve as the primary focus for most functional assays and drug design efforts. To ensure rigorous evaluation, our training set only includes RNA structures released before 2024, while excluding any RNAs sharing more than 80% sequence identity with the test RNAs. Both the training and test datasets are made publicly available at https://zhang-group.org/DRfold2/data.zip and https://doi.org/10.5281/zenodo.18401691.

## DRfold2 outperforms existing methods in global RNA tertiary structure prediction

We first evaluate the performance of DRfold2 in comparison with five state-of-the-art methods: RNAComposer [12], trRosettaRNA [17], RhoFold [18], RoseTTAFoldNA [19], and DeepFoldRNA [16]. While RNAComposer utilizes a fragment assembly and refinement approach, the other four methods are based on recently developed deep learning models for RNA structure prediction, each with different strategies and focuses. To ensure reproducibility, all methods were installed and executed locally, except for RNAComposer, which we accessed via web server. We also compare DRfold2 with the latest iteration of AlphaFold, AlphaFold3 [31], for RNA 3D structure prediction. For doing this, we submitted the RNA sequences in our test set to the AlphaFold Server and obtained the predicted structures of AlphaFold3 with default seed configurations. Fig 2A compares the TM-score and RMSD between DRfold2 and the control methods, under different sequence identity cut-off thresholds from 50% to 80% between the test and training RNAs. Here, the Template Modeling Score (TM-score) [32,33] is a length-independent scoring function to assess the quality of predicted full-length RNA structures, ranging from 0 to 1, with higher values indicating greater similarity to the native. TM-score was designed to address the limitations of the widely used Root Mean Square Deviation (RMSD), which is highly sensitive to local structure derivations. Both TM-score and RMSD are computed on the P-atoms of the predicted and experimental structures.

Within the test dataset, there are 9, 21, 37, and 41 targets with sequence identities below 50%, 60%, 70%, and 80%, respectively, relative to the training dataset. In general, the prediction accuracy of DRfold2 increases for the targets with higher sequence identity to the training set. Notably, this trend holds for all control methods even though no filters were applied to their training dataset, probably because these methods were trained on RNAs released before 2024 which have a similar homologous distance to the new test dataset used in this study. It is observed that DRfold2 consistently achieves the highest average TM-score among all methods across all sequence identity thresholds. For example, at the 80% cutoff, DRfold2 attains an average TM-score of 0.350 which is 19.9% higher than DeepFoldRNA (TM-score = 0.292). The difference is statistically significant, with a $p$-value of 6.37E−04 in the one-tailed Student $t$ test, despite the low degree of freedom, i.e., the limited sample size of 41. Even with the most stringent test subset containing 9 test targets (50% sequence identity cutoff), DRfold2 achieves an average TM-score of 0.275, which is 20.0% higher than RoseTTAFoldNA (TM-score = 0.229), with a $p$-value of 1.43E−02 (one-tailed $t$ test). Meanwhile, DRfold2 exhibits a lower average RMSD than all other control methods across different sequence identity thresholds. DRfold2 also achieves a slightly higher TM-score (0.350) and lower RMSD (14.40 Å) than AlphaFold3 (0.347 and 16.00 Å) (S1 Table); however, the difference between the two is not statistically significant, with $p$-values of 8.91E−01 and 1.37E−01 for TM-score and RMSD, respectively, in a two-tailed Student $t$ test.

Fig 2B further shows a comprehensive performance comparison using accumulative fraction plots of TM-score and RMSD. Each curve represents the fraction of predictions achieved better than specific TM-score or RMSD thresholds. DRfold2 consistently outperforms other methods in different accuracy ranges and across different sequence identity cut-offs, especially on TM-scores that are less sensitive than local fluctuations. These systematic comparison results suggest that DRfold2 provides robust global tertiary structure predictions across diverse RNA sequences and accuracy ranges.

In Fig 2C, we present one example of a CPEB3 HDV-like ribozyme from Chimpanzee (PDB ID: 7QR3), a 69-nucleotide RNA molecule, where DRfold2 shows advantage in modeling tertiary structure. Among the four better-performing methods in our benchmark, DRfold2 accurately captures the topology of the ribozyme with a TM-score and RMSD of 0.588 and 2.72 Å, respectively. In contrast, DeepFoldRNA and AlphaFold3, while capturing the general helical arrangement, show

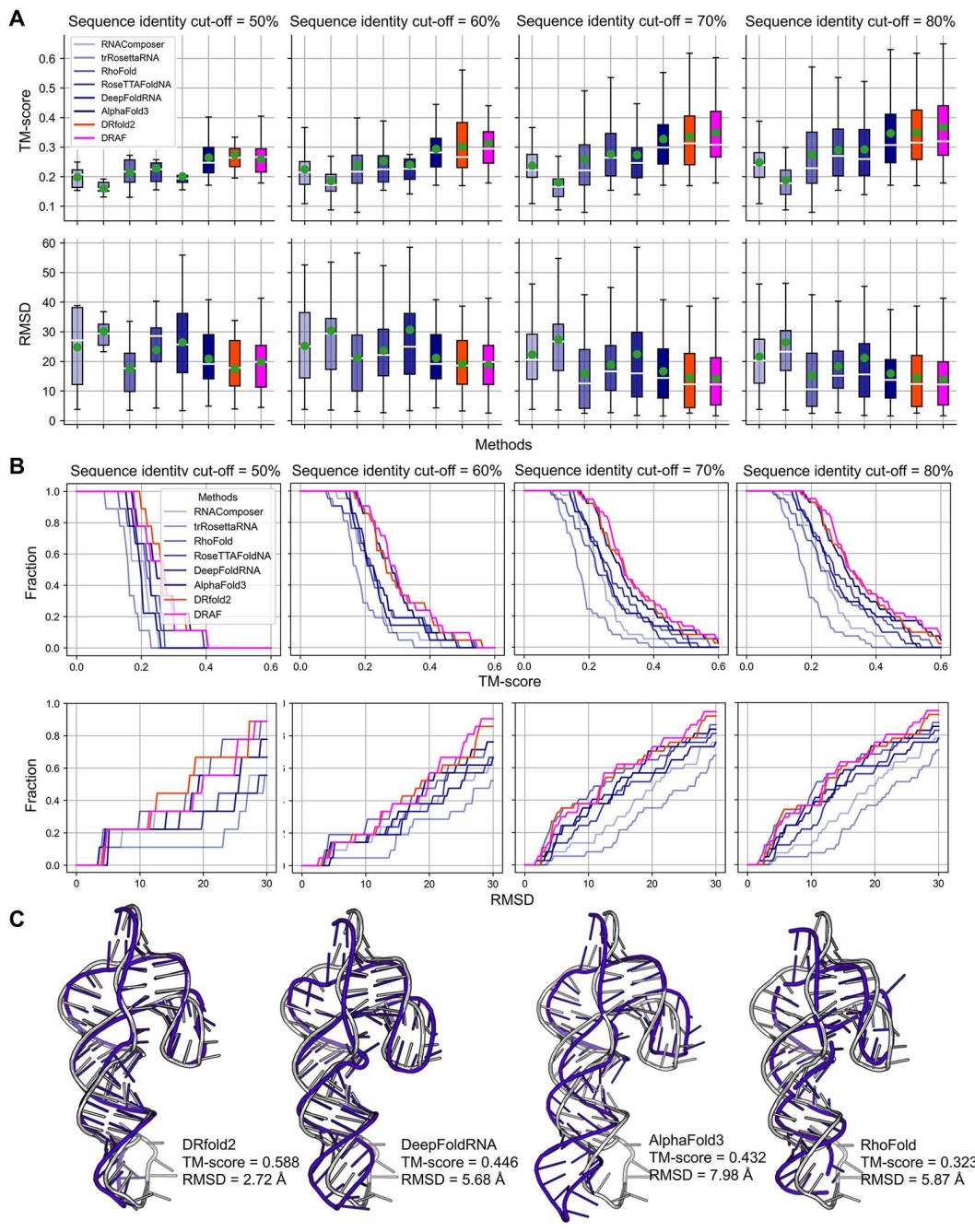

**Fig 2. RNA structure prediction results on the test dataset by different methods across different sequence identity thresholds. (A)** Box plots of TM-score and RMSD for eight RNA structure prediction methods evaluated at different sequence identity cutoffs (50%–80%) between training and test datasets, with green point and white horizontal line representing the mean and median values respectively. **(B)** Cumulative distribution plots of TM-score and RMSD for the predicted models. Upper panels display the fraction of models above different TM-score thresholds, while lower panels show the fraction of models below different RMSD values. **(C)** A representative modeling example from Chimpanzee CPEB3 HDV-like ribozyme (PDB ID: 7QR3), with models predicted by four better-performing methods (blue cartoons) overlaid on experimental structure (gray cartoons). Underlying numerical data for this figure can be found in S1 Data (see sheets "S1_Data_Figure2_A-B").

notable deviations in the orientation of the hairpin loop. As a result, the RMSD becomes 5.68 and 7.98 Å, respectively, which is at least twice the deviation of DRfold2. RhoFold shows errors in predicting the spatial organization of the junction regions, with TM-scores reducing to 0.323. Note that the maximum sequence identity between this target and the training RNAs is only 60.9%, demonstrating that the DRfold2 pipeline can provide reasonable structure predictions even for novel sequences that are not homologous to the training data.

Fig 3A and 3B provide head-to-head TM-score and RMSD comparisons by the DRfold2 and AlphaFold3. Although DRfold2 outperforms AlphaFold3 in several low- to medium-quality models, AlphaFold3 excels in two targets with

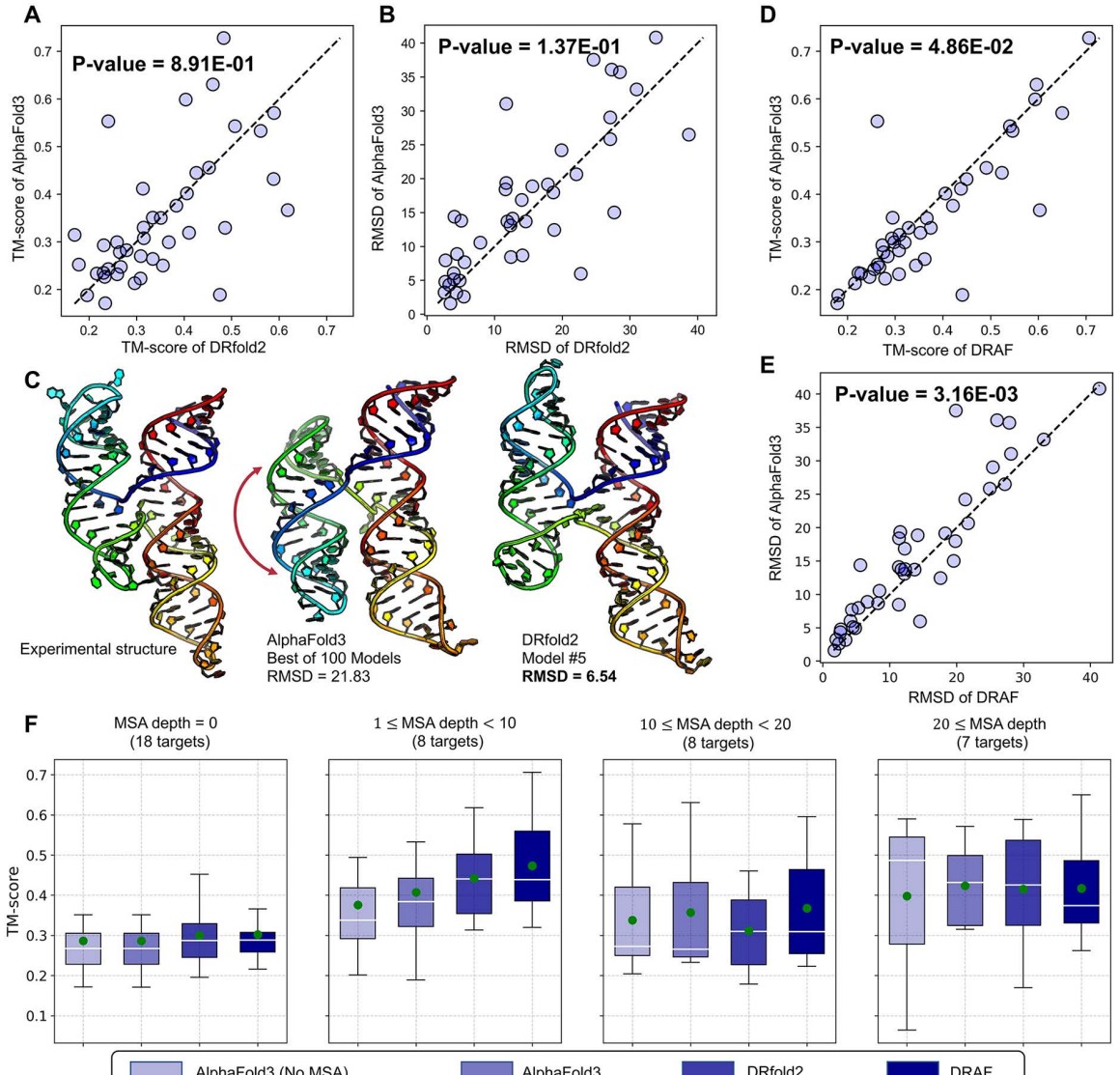

**Fig 3. Comparative analysis of DRfold2 and AlphaFold3 for RNA structure predictions. (A, B)** Head-to-head TM-score and RMSD comparisons between DRfold2 and AlphaFold3. **(C)** Structural visualization of the example from coxsackievirus B3 cloverleaf RNA (PDBID: 8DP3), showing experimental structure, AlphaFold3's best prediction from 100 models, and 5th model of DRfold2, respectively. Structures are rainbow-colored from 5′ (blue) to 3′ (red) end. **(D, E)** TM-score and RMSD comparisons between AlphaFold3 and DRfold2 with DRfold2+AlphaFold3 potential. **(F)** Performance comparisons between DRfold2 and Alphafold3 grouped by MSA depth retrieved by AlphaFold3. Green points indicate means and white horizontal lines show medians. Underlying numerical data for this figure can be found in S1 Data (see sheets "S1_Data_Figure3_A-B" and "S1_Data_Figure3_D-F").

TM-score above 0.6. A detailed comparison between DRfold2 and AlphaFold3 across different target annotations is shown in S2 Table, where pseudoknots were annotated using RNApdbee [34] and all other categories follow the official NAKB annotations [35]. DRfold2 achieves a higher average TM-score on targets containing pseudoknots, suggesting that it produces more accurate global folds in these cases. However, in terms of pseudoknot recovery rate, DRfold2 is notably lower than AlphaFold3, i.e., 56.3% versus 79.1%, indicating that AlphaFold3 captures atomic-level base-pairing conformations more accurately. For reference, other control methods have substantially lower pseudoknot recovery rates, i.e., 19.0% and 2.0% for RoseTTAFoldNA and RhoFold, respectively. For the remaining NAKB-defined categories, DRfold2 performs better on Aptamer, Catalytic, and G-quadruplex RNAs, whereas AlphaFold3 shows advantages on Virus and Riboswitch targets.

While most of our above discussions focus on the top-rank models by DRfold2, we have identified several targets for which DRfold2 generated correct structures, but the post-process step failed to select them as the first-rank models. Fig 3C illustrates one such example from the coxsackievirus B3 cloverleaf RNA (PDB ID: 8DP3), which contains 90 residues forming a 4-way junction fold [36]. By checking the first-rank models, both DRfold2 and AlphaFold3 failed to correctly predict the direction of the 4-way junction, with RMSDs to the native around 20 Å. To further explore AlphaFold3's capabilities, we submitted the same sequence to AlphaFold Server 20 times with different seeds, resulting in 100 predicted conformations. The middle panel of Fig 3C shows the best conformation with the highest TM-score, which still exhibits a poor RMSD of 21.83 Å due to incorrect chain winding topology across the 4-way junction. In contrast, the 5th-ranked model from DRfold2 successfully predicted the junction's direction, reducing the RMSD from 22.05 Å (first DRfold2 model) to 6.54 Å (right panel of Fig 3C). While this example highlights DRfold2's capability to generate diverse and alternative conformations, it also underscores the need to improve the post-processing CSOR protocol for more effectively selecting and ranking the correct models from the DRfold2 decoy pool, e.g., through the combination of third-party algorithms.

Although DRfold2 and AlphaFold3 exhibit similar overall performance, the results in Fig 3A and 3B highlight the strong complementarity between the two programs, particularly in cases where predictions deviate from the diagonal line. This complementary behavior, along with the identified need to improve DRfold2's model selection strategy, motivated us to explore whether combining the two approaches could lead to enhanced structure predictions. The optimization module in DRfold2 readily facilitates this integration by incorporating AlphaFold3 conformations as an additional potential term. Here, AlphaFold3 is used purely as an external source of candidate conformations for folding potential construction, without transferring model parameters or learned representations. No joint training, fine-tuning, or parameter sharing is performed between the two models. Guided with this combined potential function, the optimized structures achieve higher quality models than those generated by either method alone, as shown in Fig 2 labeled as DRAF, and in the detailed head-to-head comparison in Fig 3D and 3E. The hybrid approach achieves an average TM-score of 0.368 and RMSD of 14.02 Å, both being significantly better than that of AlphaFold3 (0.347 and 16.00 Å, with $p$-value = 4.86E−02 for TM-score and 3.16E-03 for RMSD, one-tailed $t$ test). These results not only validate our hypothesis regarding the complementarity between DRfold2 and AlphaFold3 but also demonstrate the flexibility and extensibility of the CSOR module, enabling the seamless integration of external model features to further improve structure prediction in DRfold2.

More strictly, we further analyzed the full-atom RMSD of control methods and DRfold2 (S1 Fig). DRfold2 consistently outperforms all control methods, achieving an average full-atom RMSD of 14.026 Å. Incorporating the AlphaFold3 potential further reduces the full-atom RMSD to 13.767 Å, highlighting the robustness of DRfold2 in generating accurate all-atom conformations.

We acknowledge that the highest TM-score among all methods, including DRfold2, is relatively low (0.368, achieved by DRfold2 combined with AlphaFold3 potentials), which may raise concerns about whether these methods can reliably predict RNA structures. We thus analyzed the number of targets according to two TM-score thresholds, 0.21 and 0.45. The lower threshold (0.21) corresponds to the expected normalized TM-score between any two random structures, while the higher threshold (0.45) indicates that two conformations belong to the same fold [32]. Using these criteria, DRfold2

successfully predicts 10 targets with the correct structural fold (TM-score ≥ 0.45) and only 3 targets below the random threshold (TM-score < 0.21). This performance is slightly better than AlphaFold3 predictions (8 and 3, respectively) and largely outperforms the non-deep learning method, RNAComposer, which fails to predict any correct folds and produces 13 structures worse than random. Considering the very low failure rate (TM-score < 0.21) of DRfold2 across different sequence identity thresholds, i.e., 7.31%, 8.11%, 14.29% and 11.11% for identity thresholds of 0.8 to 0.5, respectively, we conclude that despite the modest absolute TM-scores, DRfold2 can reliably capture meaningful overall structural topology for the majority of RNA targets, even in cases of low sequence homology.

## DRfold2 enables robust RNA structure prediction for targets with novel folds

While DRfold2 shows highly competitive performance against existing top-performing methods such as AlphaFold3, it is also of interest to examine how DRfold2 performs on targets whose structural folds differ from those present in the training set. We utilized US-align [37] to filter out targets that have maximum normalized TM-score > 0.45 against all training samples. This procedure yields a total of 25 novel-fold targets. We note that, while structurally homologous folds are excluded by this procedure, homologous sequences or closely related RNA families may still exist in the large-scale sequence corpus (RNAcentral, release 22) used for language model pretraining, which is unavoidable in self-supervised sequence modeling. However, no experimentally resolved structural information from these sequences is used during training, and the novel-fold analysis therefore does not imply reliance on trivial structural memorization.

On these novel folds, DRfold2 achieves an average TM-score of 0.309, which is lower than its performance on the full benchmark set (0.350), indicating that prediction accuracy decreases for unseen structural topologies. Importantly, DRfold2 is trained to learn a mapping from sequence to structure, and for targets with distinct sequences, the model cannot rely on simple memorization of previously observed structures to produce correct predictions. The generally higher performance of DRfold2 on those known-fold targets over novel targets, actually demonstrates that the DRfold2 model can capture latent structural representations that go beyond sequence similarity.

Notably, DRfold2 consistently outperforms AlphaFold3 on this subset of new-fold RNAs (0.309 versus 0.281), with a *p*-value of 5.29E−02 for one-tailed *t* test (S2 Fig). Furthermore, DRfold2 achieves a significantly lower RMSD (15.888 versus 18.832) with a *p*-value of 1.97E−02 (one-tailed *t* test), despite the limited number of samples. These results suggest that DRfold2 can better retain nontrivial generalization ability by extracting structural principles from sequence and does not simply memorize previously seen folds or sequences.

## DRfold2 enables robust RNA structure prediction with few homologs

Next, we examine how the performance of DRfold2 varies across different levels of evolutionary information, as reflected by the MSA depth retrieved by AlphaFold3. In Fig 3F, we present this comparison together with AlphaFold3. For reference, we also include the performance of DRAF as well as that of AlphaFold3 without MSA input. In general, both DRfold2 and AlphaFold3 achieve better performance with the increase of numbers of homologous sequences, likely due to richer co-evolutionary constraints. When no MSA is available, AlphaFold3 achieves an average TM-score of only 0.333, which is moderately lower than its full-version performance (0.347).

Although DRfold2 and AlphaFold3 show comparable overall accuracy, their behavior differs across MSA-depth groups. For targets without any detectable homologs (MSA depth = 0), DRfold2 achieves slightly higher accuracy (TM-score = 0.300) compared to AlphaFold3 (TM-score = 0.287). A similar trend is observed for low-homology targets (MSA depth 1–10), where DRfold2 outperforms AlphaFold3 with TM-scores of 0.442 versus 0.407. For targets with richer evolutionary information (for example, MSA depth 11–20), however, AlphaFold3 performs better (TM-score = 0.357 versus 0.311 for DRfold2). Nonetheless, the hybrid folding pipeline, DRAF, which integrates DRfold2 predictions with the AlphaFold3 potential, achieves the highest accuracy (TM-score = 0.368) in this group, demonstrating the complementarity of the two approaches.

Taking together, these results further highlight the complementary strengths of DRfold2 and AlphaFold3. DRfold2 is trained primarily on single-sequence features enriched with co-evolutionary signals derived from RCLM. The hidden universal covariance learned by RCLM becomes more valuable, for example, when explicit homologs are insufficient or not detected. Consequently, DRfold2 can provide more reliable predictions for RNAs with few or even no homologs.

## DRfold2 produces physically realistic RNA structures

In addition to achieving high global structure accuracy, DRfold2 also produces physically correct models, addressing a crucial problem in current RNA bioinformatics [38]. DRfold2 employs a hybrid refinement strategy that integrates Monte Carlo simulation with L-BFGS optimization to generate the final full-atomic models. We evaluated the geometric quality of the refined structures on the test set using three standard metrics, i.e., the root mean square (RMS) of bond length and bond angle deviations, and the clash score (S3 Table). The corresponding values for native structures are also provided as references. The refinement procedure substantially improves all three metrics, reducing them from 0.042, 5.854, and 42.823 to 0.012, 1.559, and 2.171, respectively. Notably, DRfold2's refinement step yields structures with an average clash score even lower than that of the experimental structures, highlighting its strong physical regularization capability.

S3 Table further reports the number of structures containing at least one entanglement [39], as identified by RNAspider, a webserver for detecting topological entanglements in RNA 3D structures [40]. Interestingly, although DRfold2 refinement is not explicitly designed to resolve such motifs, the refinement successfully removes one D&L interlace, in which a dinucleotide step (D) and a loop (L) are interwoven (S3 Fig).

We further examined the local torsion-angle similarity by evaluating the Mean of Circular Quantities (MCQ), a torsion-based distance metric that assesses the local geometry defined by every consecutive four atoms, with a lower MCQ value indicating better local structure [41]. S4 Fig presents the MCQ comparison across all baseline methods and DRfold2. On average, DRfold2 achieves an MCQ of 23.8, significantly outperforming many control methods (RNAComposer, trRosettaRNA, and RhoFold). For example, RNAComposer, which assembles structures using a high-quality fragment library and thus can generally provide good local torsion geometry, still shows a higher MCQ of 26.1 ($p$-value = 1.06E−03, one-tailed $t$ test). The MCQ value of DRFold2 is, however, comparable or slightly higher than the rest of the control methods (RoseTTAFoldNA, DeepFoldRNA, AlphaFold3), suggesting potential future improvements, for example, through the incorporation of torsion-based energy potential.

In summary, DRfold2 provides not only accurate global folds but also physically well-behaved structures, ensuring that the predicted models are suitable for reliable downstream experimental design and computational analyses.

## DRfold2 networks enhance the accuracy of RNA secondary structure prediction

Many RNA functions are closely associated with their secondary structure arrangements, which are primarily stabilized by hydrogen bonding and base-pair stacking interactions [42]. Fig 4A–4D analyzes the Interaction Network Fidelity (INF) indices, which are defined as the geometric mean of precision and recall of secondary structure predictions, i.e., $\text{INF} = \sqrt{\text{precision} \times \text{recall}}$. First, for Watson-Crick base pairs (INF_wc) and stacking interactions (INF_stack), while most methods (except for trRosettaRNA) demonstrate reasonable performance, DRfold2 achieved the INF values of 0.831 and 0.754, slightly lower than those of AlphaFold3 (0.849 and 0.762), but higher than all other control methods. For non-canonical interactions (INF_nwc), however, all methods, including DRfold2 and AlphaFold3, exhibit much lower INF values. Although DRfold2 and AlphaFold3 still achieves the top-2 INF_nwc, their low value (0.184 and 0.202, respectively) underscores that accurately modeling non-canonical base-pair interactions remains an urgent challenge in RNA structure prediction. Fig 4D presents a combined secondary structure accuracy across all three indices. DRfold2 achieves INF_all = 0.750, which is statistically comparable to that of AlphaFold3 ($p$-value = 2.92E−01, two-tailed $t$ test), and 7.4% higher ($p$-value = 5.97E−04, one-tailed $t$ test) than $t$he third-best method, DeepFoldRNA (INF_all = 0.698).

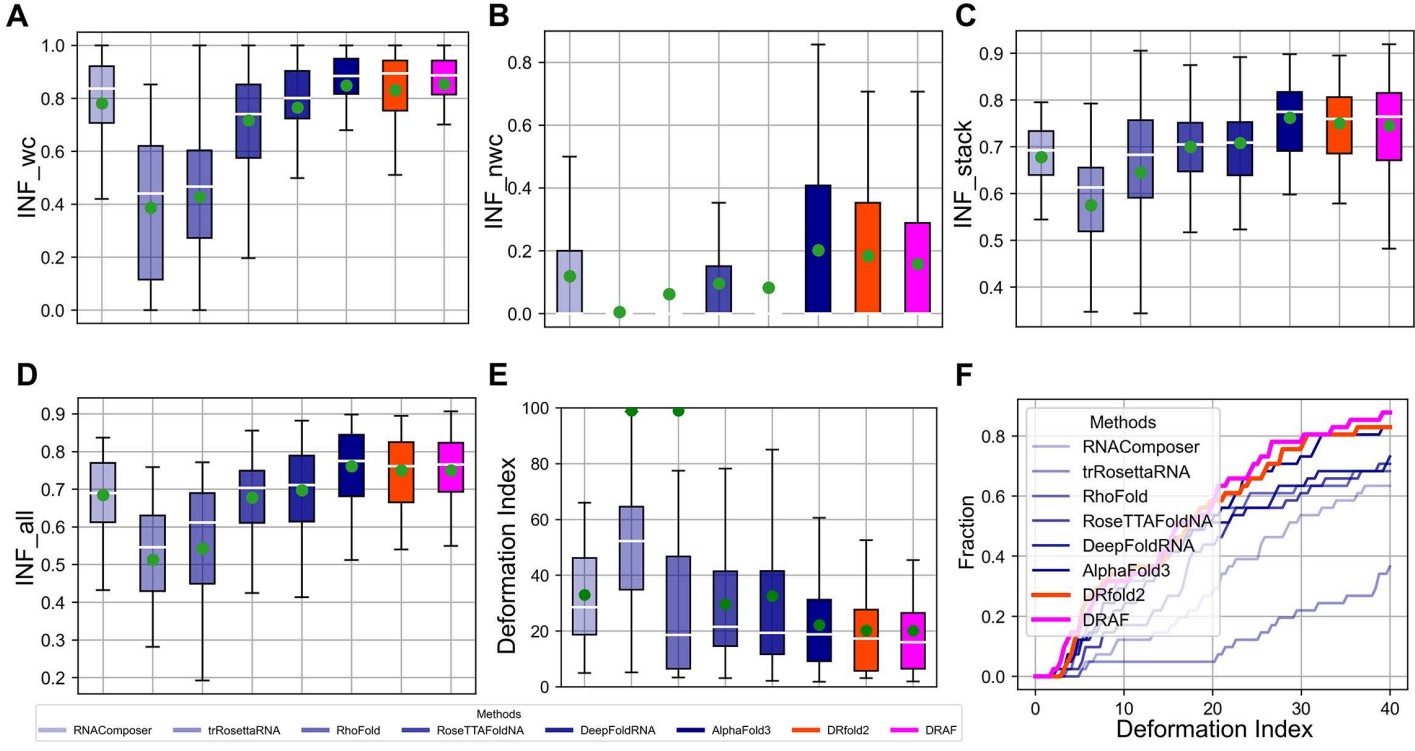

**Fig 4. Comparative analysis of RNA secondary structure prediction. (A–E)** Box plots comparing eight RNA structure prediction methods across different RNA secondary structure performance metrics. The green point and the white horizontal line represent the mean and median values respectively. Note that the mean Deformation Indexes for trRosettaRNA and RhoFold are much higher than the median, due to their poor performance on a few targets where no correctly modeled base pairs are present in the predicted structures. **(F)** Cumulative distribution plot of deformation index for different methods. Underlying numerical data for this figure can be found in S1 Data (see sheets "S1_Data_Figure4_A-F").

While the INF indices assesses the conformational accuracy at the nucleotide pair level, the Deformation Index (DI), defined as RMSD/INF_all, has been introduced to provide a more balanced evaluation of 3D topological and 2D base-pairing patterns [43]. As shown in Fig 4E, DRfold2 achieves the lowest average DI of 20.21, which is 9.6% lower than the second-best method, AlphaFold3 (DI = 22.15), The corresponding p-value of 1.43E-01 in two-tailed *t* test indicates *t*hat this improvement is not statistically significant. However, the method that combines DRfold2 and AlphaFold3 potentials (DRAF) achieves the lowest DI of 20.15, which is significantly lower than that of AlphaFold3 (p-value = 2.14E−02, one-tailed *t* test). Fig 4F further presen*t*s the accumulative fraction distribution at different DI cut-offs, where DRfold2 outperforms all third-party control methods, achieving the highest normalized area under the curve (AUC = 0.520), which is 5.3% higher than the best control method, AlphaFold3 (AUC = 0.494). These results suggest that DRfold2 excels not only in predicting global structural topology but also in capturing detailed hydrogen-bonding and base-pair stacking patterns within RNA structures.

It is notable that DRfold2 has achieved superior secondary structure predictions using sequence information alone, without incorporating any precomputed secondary structure predictions, as many control methods do. This success can be attributed to several key components of DRfold2. First, the RCLM effectively captures inter-residue co-evolutionary patterns, providing reliable embedding features for the deep learning model. Second, the Denoising Structure Module helps enhance the end-to-end learning of inter-nucleotide frames through supervised training, ensuring proper spatial relationships between nucleotides. Finally, the precise prediction of inter-nucleotide geometries through multi-task learning helps guide the refinement of base pairing conformations during the post-processing optimization stage.

## RCLM more effectively captures co-evolutionary patterns than existing language models

One of the main innovations of the DRfold2 pipeline is the introduction of RCLM which is expected to provide a more precise approximation of the full likelihood probability by considering higher-order inter-nucleotide interactions (see Materials and methods). To examine the assumption, first, S5 Fig evaluates the basic nucleotide sequence recovery rate of RCLM against three control language model (LM) methods, i.e., RNA-FM, RiNALMo, and RNAErnie. Here, RNA-FM is a representative RNA sequence language model [44] trained on RNAcentral database [45] with conventional Masked Language Model Loss. RiNALMo was trained with 650M parameters on 36M non-coding RNA sequences from several database [46]. RNAErnie was trained through a specifically designed motif-level random masking strategy [47]. During the training, the composite likelihood of RCLM was calculated based on the sum of two probabilities for nucleotide-wise masked type and pair-wise joint predictions. In S5 Fig, we also examine the sequence recovery ability derived from the pair-wise joint probability. Here, we marginalize the predicted joint distribution into nucleotide-wise distribution for evaluation, denoted as RCLM-M.

The data show an improved masked nucleotide recovery ability for RCLM, with a recovery rate of 45.82%, compared to RNA-FM (41.47%) and RNAErnie (42.87%). Interestingly, the marginalized prediction (RCLM-M) achieves slightly better performance, with a sequence recovery rate of 47.87%, marginally higher than the second-best model, RiNALMo. Notably, these evaluation metrics are marginalized across positions, representing only a subset of RCLM. Even with this partial probability, RCLM demonstrates consistently superior performance over RNA-FM. The result of RCLM-M suggests that RCLM also generates meaningful predictions for the pair-wise joint nucleotide distributions.

In addition to sequence recovery rate, which is highly related to the masking and loss function strategy during the training and cannot fully capture the performance of the model, a more critical assessment is on the co-evolution patterns captured by the language models, which are directly relevant to 3D RNA structure predictions. To evaluate this capability, we employed a completely unsupervised method, Categorical Jacobian [48], to extract the RNA contact maps from the language models. Fig 5A–5C presents a comparison of top-$N$ contact map precision between RCLM and control LMs across distance thresholds of 12, 16, and 20 Å, where RCLM consistently has dramatically higher precisions than all control language models. For example, when using a contact distance cutoff of 12 Å, the Top-$L$ precision of RCLM (49.0%) is more than 100% higher than that achieved by RNA-FM (23.6%). RCLM also produces significantly higher contact prediction than the second-best LM model, RiNALMo, with $p$-values of 4.82E−09, 5.00E−09, and 1.04E−09 for thresholds of 12, 16, and 20 Å, respectively, in one-tailed $t$ test. In Fig 5A–5D, we further examine the precisions of the Categorical Jacobian contact and secondary structure predictions based on marginalized pairwise prediction from RCLM (RCLM-M). While RCLM-M produces slightly lower precisions than RCLM, it still maintains a large performance margin compared to those control methods. In Fig 5D, we also compare the precision of unsupervised secondary structure prediction, by assuming that the Categorical Jacobian produces the base pairing score maps. RCLM, RCLM-M, and RiNALMo exhibit comparable performance, while RCLM once again achieves highest precision and shows a statistically significant advantage over the second-highest model, RiNALMo, in Top-$L/2$ precision, with $p$-value of 8.57E-03 in one-tailed $t$ test. These results indicate that RCLM can more effectively learn high-quality co-evolution patterns from sequence data than existing language modeling approaches. Since Categorical Jacobian requires mutation of all positions into all possible nucleotides during the extraction process, the high contact precision by RCLM also suggests that the Composite Language Model has achieved a reliable level of sensitivity. It is important to note that, RCLM, containing 47M parameters, demonstrates superior performance in unsupervised contact precision tasks, with only 50% of the model size of RNA-FM (100M) and RNAErnie (87M), and even less than 10% of RiNALMo (650M). Such efficiency makes this form of language model training a promising approach for RNA sequence language model learning. Our next step is to upscale this methodology to explore the full potential of RCLM.

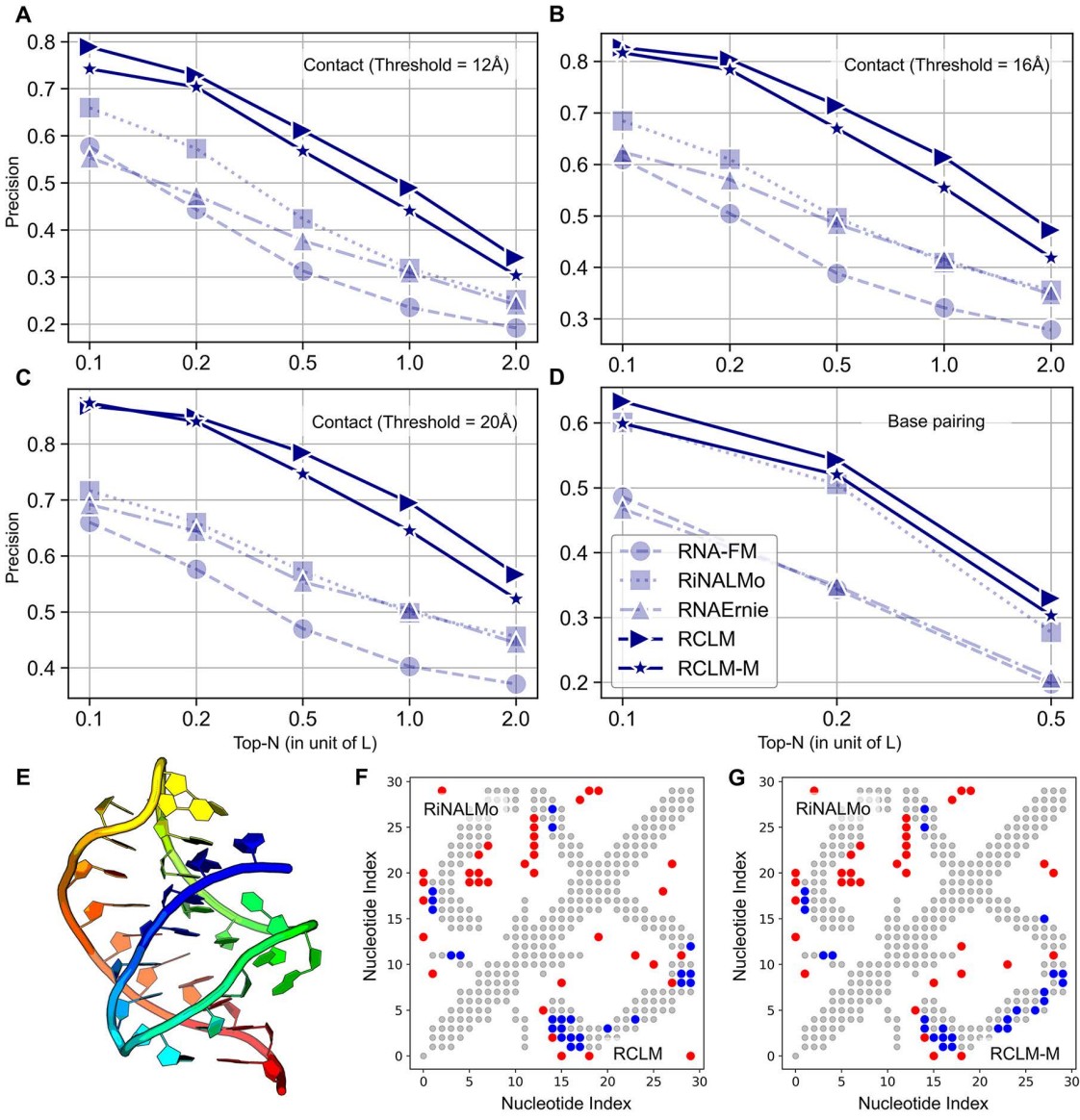

**Fig 5. Comparative analysis of RCLM and control RNA language models for RNA sequence learning. (A–C)** Top-*N* precision curves at different contact thresholds (12, 16, and 20 Å) for RCLM, RCLM-M, and control language models. **(D)** Precision comparison of unsupervised RNA secondary prediction between RCLM, RCLM-M, and control language models. **(E)** 3D structure visualization of the Class I type III preQ1 riboswitch from E. coli (PDB ID: 8FZA). **(F, G)** Contact maps of 8FZA, with gray dots representing ground-truth contacts, red dots false predicted contacts, and blue dots true positive predictions. Upper-left triangle shows results of RiNALMo predictions, and lower-right triangle is that of RCLM (**F**) and RCLM-M (**G**) predictions. Underlying numerical data for this figure can be found in S1 Data (see sheets "S1_Data_Figure5_A-D" and "S1_Data_Figure5_F-G").

As an illustration, Fig 5E–5G showcase an example of a Class I type III preQ1 riboswitch from *Escherichia coli* (PDB ID: 8FZA) [49], which consists of 30 nucleotides as shown in Fig 5E. RCLM and RCLM-M achieve reasonable predictions on this target with Top-*L* contact precision of 56.67% and 63.33%, respectively, where the contacts are defined by inter-N atom distances <12 Å. The second-highest precision among the control methods is 23.33% from RiNALMo. The upper-left and lower-right panels of Fig 5F and 5G are the unsupervised Top-*L* predicted contacts by RiNALMo and RCLM/RCLM-M, respectively, where the ground-truth contacts are shown as gray dots. RiNALMo only predicts 7 out of 30 (23.33%)

positive contacts, while RCLM and RCLM-M correctly produce 17 (56.67%) and 19 (63.33%) positive contacts when using single nucleotide prediction and marginalized pairwise prediction, respectively, both exceeding RiNALMo's accuracy by more than 2-fold. The result of this example highlights again the efficiency of RCLM to learn co-evolution information that is crucial for producing higher-precision structural restraints and ab initio RNA models, especially for the cases with limited sequence and structural homologies.

## RCLM learns base pairing interactions in an unsupervised manner

Although the model is optimized purely for likelihood-based sequence reconstruction, the coevolutionary statistics it captures are validated to correlate strongly with physical contacts (Fig 5). To further understand how RCLM learns these physical interactions, we conducted a nucleotide-masking experiment (in addition to the Categorical Jacobian analysis). Specifically, for each annotated base-pairing interaction, we masked one nucleotide and assessed how accurately RCLM could recover the masked residue from the remaining context, and the corresponding attention weight distributions. Representative examples are shown in S6 Fig for two targets whose PDB IDs are 8HZD (chain A) and 8Z8U (chain A), respectively. In 8HZD_A, masking positions 2 or 6 causes the attention mechanism to assign higher weights to their long-range canonical and non-canonical partners, indicating that RCLM learns these interactions by attending directly to the physically paired nucleotides. A similar behavior is observed in 8Z8U_A for positions 1 and 20. However, this physically grounded attention does not always generalize, especially for some non-canonical interactions. For instance, when masking position 8 in 8HZD_A or position 13 in 8Z8U_A, the model relies more on local neighboring nucleotides rather than attending to the true long-range interaction partner.

Overall, for canonical Watson–Crick pairs, the recovery success rate was 57.1%, indicating that RCLM effectively captures canonical interacting patterns. However, for non-canonical interactions, the recovery rate dropped to 36.6%, suggesting that RCLM learns weaker signals for these interaction types (S7 Fig). Both canonical and non-canonical base pairs are recovered at higher accuracies above random expectation (~25%), indicating that RCLM has learned to recognize true physical pairing signals without any structural supervision. Such difference in accuracy may partly account for the higher performance of DRfold2 on canonical, compared to non-canonical, base-pair modeling (Fig 4B).

We further performed a two-nucleotide masking experiment on annotated non-canonical pairs. In this setting, both nucleotides in a non-canonical interaction were masked, and we evaluated whether RCLM could correctly recover both nucleotide identities using the sequence-level prediction head and the pairwise prediction head (supervised by the pairwise likelihood term). Overall, the pairwise head achieved a slightly higher recovery rate (18.0%) compared with the sequence head (17.8%). The pairwise head outperformed the sequence head in 9 targets, while the sequence head outperformed the pairwise head in 5 targets. However, the difference is not statistically significant ($p$-value$=7.81E-01$, two-tailed $t$ test), indicating that the composite likelihood does not provide a significant advantage for non-canonical interactions. This observation, however, aligns with our expectation since all nucleotide pairs are treated equally without bias terms for non-canonical pairs during the RCLM training.

In summary, these results confirm that, while RCLM provides strong support for canonical base-pair learning, its ability to model non-canonical interactions remains limited. Nevertheless, they also suggest potential directions for improvement, such as explicitly modeling non-canonical interactions in composite likelihood.

## Positive impact of Composite Language Model on RNA 3D structure prediction

To directly examine the effectiveness and impact of different language models on 3D RNA structure prediction, we present in Fig 6A the TM-scores of identical end-to-end structural learning networks built on one-hot and RCLM representations. We also include RNA-FM as a reference model, which was previously shown to be helpful for RNA structure prediction [50]. It is observed that models using RCLM features achieve the highest mean and median TM-scores of 0.319 and 0.299, which are 6.0% and 10.7% higher than that with RNA-FM (0.301 and 0.270) and 7.8% and 12.4% higher than that

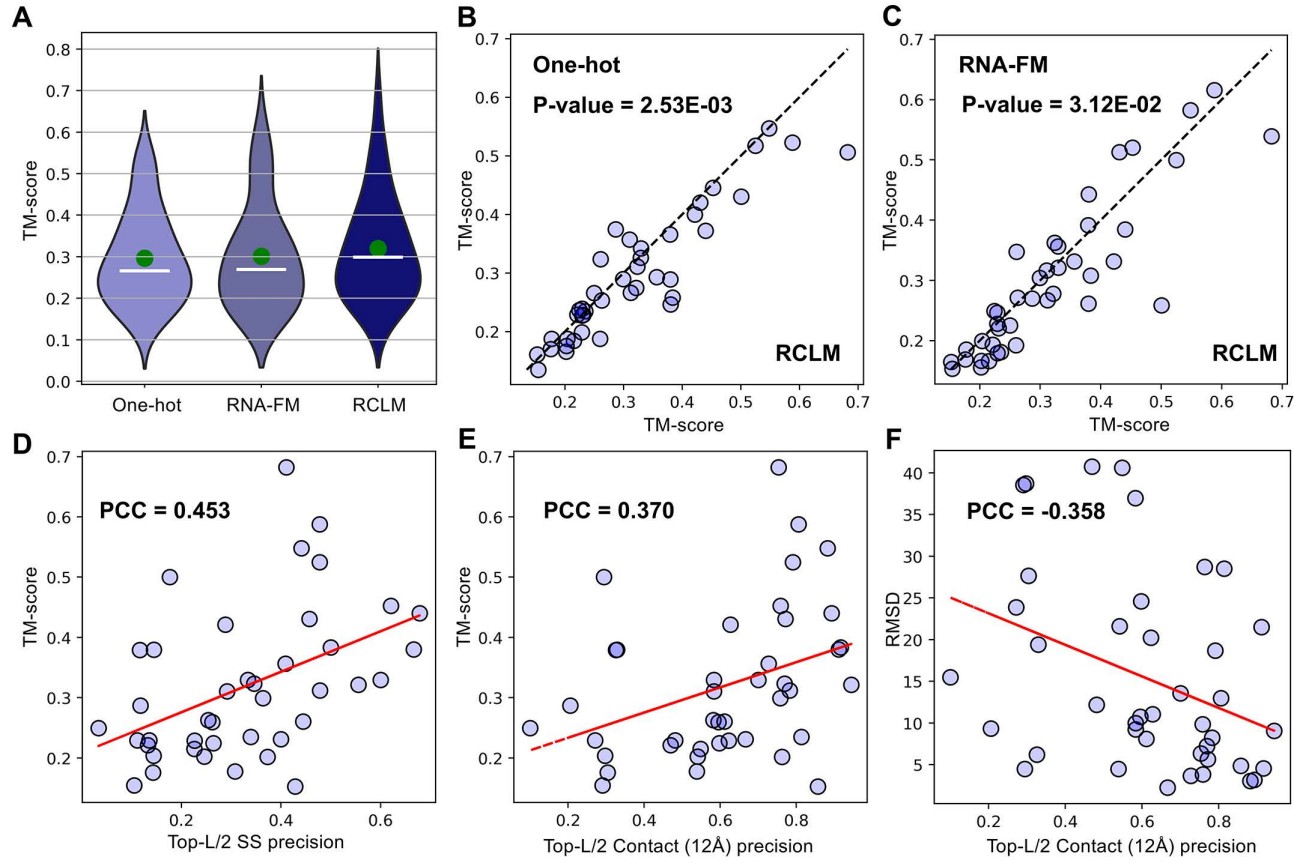

**Fig 6. Comparison of RNA structure prediction performance across different encoding methods and correlation analysis between unsupervised contact/secondary structure precision and TM-score/RMSD. (A)** Violin plots of TM-scores by end-to-end RNA structure predictions based on different encoding strategies (One-hot, RNA-FM, and RCLM), with white horizontal lines indicating medians and green points showing means. **(B, C)** Head-to-head TM-score comparisons between RCLM and One-hot and RNA-FM encoding based models. **(D)** Correlation between TM-score and Top-$L/2$ secondary structure precision. **(E)** Correlation between TM-score and Top-$L/2$ Contact precision at 12Å threshold. **(F)** Correlation between RMSD and Top-$L/2$ Contact precision at 12Å threshold. Red lines indicate linear regression fits. Underlying numerical data for this figure can be found in S1 Data (see sheets "S1_Data_Figure6_A-C" and "S1_Data_Figure6_D-F").

of the one-hot representation (0.296 and 0.266), respectively. Fig 6B and 6C further list head-to-head TM-score comparisons of RCLM against the control models, where RCLM outperforms one-hot and RNA-FM in 30 and 25 cases, while underperforming in only 11 and 16 cases, respectively.

To quantitatively interpret the contribution of language models to the 3D RNA structure modeling, Fig 6D shows the relationship between the TM-score of RCLM-based models and the Top-$L/2$ secondary structure precision from RCLM Categorical Jacobian, where a medium-high Pearson Correlation Coefficient (PCC = 0.453) is observed, suggesting that targets with a higher RCLM-based secondary structure precision generally have more accurate 3D structure prediction. In addition to base pairing precision, Fig 6E and 6F also examine the correlation between unsupervised contact map precision with TM-score and RMSD, respectively, which show slightly weaker but robust PCC values of 0.370 and −0.358. Collectively, these results suggest that the comprehensive representations provided by advanced language models, like RCLM, can enhance the ability to learn co-evolutionary patterns and spatial restraints, leading to more accurate 3D RNA structure modeling through the end-to-end network in DRfold2.

## Performance of DRfold2 in CASP16

An earlier version of DRfold2 participated in the 16th Critical Assessment of protein Structure Prediction (CASP16) experiment for RNA structure prediction under the group ID "dNAfold." S4 Table summarizes the average TM-score and accumulated Z-scores for the Top 20 groups on the 20 RNA targets with known experimental structure and sequence lengths below 400 nucleotides. Although the CASP16 assessment contains both RNA and DNA targets, only monomer RNA targets, for which dNAfold submitted predictions, were included in this analysis.

The results revealed that Vfold, which utilizes a template-based modeling approach integrated with AlphaFold3 [50], achieved the highest performance, with an average TM-score of 0.566 and Z-score of 19.71, outperforming other groups. The remaining top-ranked groups had comparable average TM-scores around 0.5, with DRfold2 securing 5th place in both TM-score (0.510) and Z-score (13.46), indicating its strong competitiveness among the state-of-the-art programs. Consistent with the benchmark results shown in Fig 3, DRfold2 slightly outperformed the AlphaFold3 server, which was ranked 7th in TM-score (0.501) and 16th in Z-score (9.78) (S4 Table).

It is noteworthy that the AlphaFold3 server was released before the CASP16 experiment, allowing the participating groups to integrate AlphaFold3 predictions into their modeling pipelines, which many did [51]. However, we chose not to incorporate AlphaFold3 or any third-party predictions in our CASP16 submissions, aiming to more critically assess the DRfold2's independent modeling capabilities. To further assess the uniqueness of the RNA structure predictions, we introduced an AlphaFold dissimilarity score, defined as $1 - TM_M$, where $TM_M$ is the maximum TM-score between the first submitted model by each group and the five models submitted by AlphaFold3 (Group ID: AF3-server). In S8 Fig, we plot the average TM-score on 20 RNA monomer targets versus the AlphaFold dissimilarity for all groups that submitted predictions for all these targets, where dNAfold exhibited the highest dissimilarity score among the six groups that have an average TM-score outperforming AlphaFold3. Moreover, dNAfold achieved the top rank among all groups with a dissimilarity score above 0.45. These results highlight the capability of DRfold2 for producing distinct and independent structure predictions. Nevertheless, given the complementary nature of DRfold2 and AlphaFold3 in both methodology and the benchmark results shown in Fig 3, we expect that appropriate integration of DRfold2 with AlphaFold3 or other top-performing models represents a promising direction for future improvements in DRfold2-based RNA structure prediction.

## DRfold2 running time analysis

The DRfold2 pipeline can be broadly divided into two stages: (1) deep learning inference for generating multiple decoys and geometry restraints, and (2) post-processing to obtain the final structure. In S9 Fig, we present a running time analysis of both the deep learning inference and the post-processing against RNA length. To avoid bias, the refinement step using RNArefine is not included in this analysis, as its runtime is fixed at 180 min.

We observe that for the majority of targets, DRfold2 can generate 80 coarse-grained conformations within 40 min, i.e., <30 s per model on a single RTX 5,000 GPU without parallelization. Even for the largest tested RNA, the maximum inference time remains <1 min per conformation. The post-processing stage is also well controlled despite the included complex optimization procedures, and its running time is approximately 0.834× that of the deep learning inference.

In summary, for typical RNA sizes (RNA length <200 nts), DRfold2 completes folding within 40 min, while larger RNAs of ~350 nucleotides can be predicted within <120 min to obtain an accurate global fold. An additional 180-min refinement step can further produce high-quality full-atomic structures with largely improved physical correctness.

## Discussion

RNA structure prediction has been largely constrained by the limited availabilities of experimental structures that can be used for template recognitions and/or training effective deep learning models. We have developed DRfold2, a new deep learning-based method for ab initio RNA structure prediction by integrating a newly trained RNA Composite Language Model with a denoising-coupled end-to-end learning strategy. The method has been tested on a non-redundant test set

with different redundancy cut-offs and shown consistently improved performance over existing approaches in both 3D tertiary structure prediction and 2D base pair modeling, demonstrating the effectiveness and robustness of the proposed method.

Although the performance of DRfold2 is primarily evaluated using coarse-grained metrics such as TM-score, we have also assessed the predicted structures using full-atom RMSD, and DRfold2 consistently outperforms all control methods. This indicates that the global topology captured by coarse-grained metrics is also well reflected at the full-atom level. While DRfold2 does not achieve the best performance for certain full-atom descriptor (MCQ), the MCQ metric, however, marginalizes local variations and assumes that all local regions contribute equally. In reality, local deviations in critical loops or junctions can have a larger impact on the overall global topology, and may not be fully captured by averaged local metrics. Other full-atom physical validation metrics, such as RMS deviations of bonds and angles or clash scores, focus primarily on geometric plausibility and the likelihood of the structure itself, but do not specifically evaluate $P$(structure|sequence), i.e., whether the predicted conformation is compatible with the RNA sequence. Well-established coarse-grained global metrics, such as TM-score and RMSD, while sometimes simpler, better reflect this sequence-informed structural prediction accuracy. Nevertheless, by combining coarse-grained global metrics with selective full-atom evaluations, our evaluation approach confirms that DRfold2 predictions are both globally accurate and physically realistic.

The key factor contributing to the success of DRfold2 is the proposed RCLM, trained from scratch with an improved loss function for better likelihood approximation. The RCLM model achieves significantly higher unsupervised contact prediction precision compared to existing representative methods using only 10% of the parameters, demonstrating a superior ability and efficiency in capture co-evolutionary signals solely from sequence data. The correlation between contact precision and structural accuracy further validates its effectiveness in learning structurally relevant patterns directly from sequences. Meanwhile, the newly introduced DRSM enhances the robustness of the end-to-end training process.

While DRfold2 shows comparable overall performance to AlphaFold3 on our test set, our analysis suggests that the two methods provide highly complementary predictions. By incorporating AlphaFold3 predictions as an additional potential term in DRfold2 optimization framework, we achieved statistically significant improvements in both TM-score and RMSD, compared to either approach.

The CASP16 results further validate the effectiveness of DRfold2, particularly for regular-sized monomer RNAs. However, we observed relatively lower performance and ranking on large RNA targets above 200 nucleotides. One possible reason is that many groups incorporated AlphaFold3, which has twice the crop size of DRfold2, whereas we intentionally evaluated DRfold2 independently, relying solely on our own method for testing. We understand that the CASP competition naturally emphasizes prediction accuracy of submitted models, which often leads participants to combine multiple existing methods for the best possible results. However, developing new independent methods remains important for the field. DRfold2 offers a distinct approach that uses the Composite Language Model, deep end-to-end learning, and post optimization protocols, providing a different perspective on RNA structure prediction, as highlighted in the AlphaFold3 dissimilarity analysis by S8 Fig. Although it may not always outperform top-performed models like AlphaFold3, the diversity in modeling approaches prevents over-dependence on a single method and creates opportunities for future innovations.

## Materials and methods

DRfold2 is new pipeline for RNA 3D structure prediction consisting of four core modules: (1) RNA Composite Language Model, (2) RNA Transformer Block, (3) Denoising Structure Module, and (4) final model selection and refinement through the CSOR Protocol (Fig 1). While it shares a similar name with the earlier DRfold method [21], DRfold2 introduces key advancements built on a completely different framework, most notably with the incorporation of the Composite Language Model, which greatly enhances RNA sequence and pairwise representations. Additionally, the pipeline integrates a DRSM, which employs a controlled perturbation strategy to robustly learn structural transformations by efficiently correcting noisy RNA conformations. The pipeline also features a carefully designed flexible end-to-end loss function

with distance-dependent reweighting schema to help the model learn meaningful interaction patterns. Lastly, we extend the deep learning-based conformational pool for the proposed CSOR to identify the most promising structure and possible alternative conformations. More detailed technical comparisons of the newly introduced modules and algorithmic upgrades are summarized in S5 Table. Equipped with those advancements, DRfold2 significantly outperforms DRfold (S10 Fig) in TM-score (0.350 versus 0.300) with *p*-value of 8.03E-04 in one-tailed *t* test, providing 2.5 times as many (from 4 to 10) foldable RNAs (TM-score > 0.45) on the test set used in this study.

### RNA sequence representation learning by Composite Language Model

The first component of DRfold2 is a pre-trained RNA language model, RCLM, which can provide enhanced RNA sequence representation. Given an input RNA sequence with $L$ nucleotides, which is represented by one-hot encoding, two types of sequence and pair representations will be generated through RCLM (Fig 1B). The initial sequential representation $s_{Rep} \in R^{L \times 512}$ can be obtained by applying a linear layer over the one-hot feature. In addition, another two separate sequence projections above $s_{Rep}$ will be added vertically and horizontally to form the initial pair embedding $z_{Rep} \in R^{L \times L \times 128}$. Here, 512 and 128 are the representative dimensions for sequence and pair features, respectively.

The initial sequential and pair representations, $s_{Rep}$ and $z_{Rep}$, will be used as the input of a set (18 in RCLM) of RNA Transformer Blocks. As shown in Fig 1C, each transformer block is designed to model pairwise interactions, which are critical for biomolecules. Here, the RNA Transformer Block used in DRfold2 is based on the Evoformer architecture from AlphaFold2, which was originally designed to model MSA data. In our setting, a single RNA sequence can be viewed as a degenerate MSA with no aligned homologs. Since the underlying block architecture remains unchanged, we preserve the original AlphaFold2 formulation (e.g., Outer Product Mean) for this component. It is worth noting that two sub-modules, i.e., Triangle Attention StartingNode and Triangle Attention EndingNode modules, were omitted in RCLM, as this configuration can reduce the GPU memory consumption.

In addition to the pair modeling, a key innovation of RCLM is the implementation of composite likelihood maximization for masked regions during the training. Given a sequence $x = (x_1, x_2, \cdots, x_L)$, the objective of masked language models is to maximize the probability of masked regions $\mathcal{M}$, i.e., $P(X_i = x_i, \forall i \in \mathcal{M})$. For notational simplicity, we omit the conditioning on unmasked regions and network parameters, though they are included in the full conditional probability. We find that existing mainstream BERT-based language models [52] approximate this distribution by maximizing a *pseudo-likelihood* $\prod_{i \in \mathcal{M}} P(X_i = x_i)$, assuming that each masked token $X_i$ is independent of the others. However, this independence assumption may not always hold, especially for RNAs, where multiple interactions occur among nucleotides.

To better approximate the actual full likelihood in RNAs, we consider maximizing a composite likelihood ($C\mathcal{L}$) [26], given by

$$C\mathcal{L} = \prod_{c \in M} P(X_c = x_c)$$

(1)

where $C$ is a set of masked position subsets. When $C = \{\{i : i \in \mathcal{M}\}\}$, $C\mathcal{L}$ corresponds to the full likelihood that models the joint distribution of all masked positions simultaneously. On the other hand, if we set $C = \{\{i\}, i \in \mathcal{M}\}$, $C\mathcal{L}$ reduces to the pseudo likelihood, where each masked token is assumed to be conditionally independent of others. In our implementation of RCLM, we incorporate second-order interactions by setting $C = \{\{i\}, i \in \mathcal{M}\} \cup \{\{i,j\}, i \in \mathcal{M} \text{ or } j \in \mathcal{M}\}$, which considers both individual tokens and interactions between pairs of masked positions. Thus, the final loss function for our RCLM training is the negative log-composite likelihood:

$$L_{RCLM} = -\left( \sum_{i \in \mathcal{M}} \log P(X_i = x_i) + \sum_{i,j:\ i \in \mathcal{M}\ OR\ j \in \mathcal{M}} \log P(X_i = x_i, X_j = x_j) \right)$$

(2)

In this case, we consider both first- and second-order interactions, which naturally align with the sequential and pair representations learned by the RNA Transformer Blocks. Notably, composite likelihood maximization has been previously applied in protein sequence modeling, such as clmDCA [53] for direct coupling analysis. Here, we incorporated pairwise composite likelihood maximization into RNA language model training, to enable the capture of complex nucleotide interactions that go beyond simple positional independence assumptions. In our setup, when position $i$ and $j$ are masked, while the pair labels $(x_i, x_j)$ and $(x_j, x_i)$ correspond to the same nucleotide pair, they are assigned to different classes based on the order of $i$ and $j$. Specifically, the joint states of a nucleotide pair are categorized into 16 classes, and $(x_i, x_j)$ and $(x_j, x_i)$ fall into distinct (though related) categories. These symmetric and asymmetric operations in Sequence Embedder and RNA Transformer Blocks allow the model to capture directional or order-dependent correlations between nucleotide positions.

The RCLM model is composed of 18 transformer blocks, with hidden dimensions of 512 for sequence representations and 128 for pairwise representations, resulting in a total of 47,495,200 trainable parameters. The model was trained on ~30M RNA sequences from the RNAcentral database [45] (Release 22) over 67,000 batches, with a batch size of 128. The entire training process took about 15 days using a single NVIDIA A40 GPU.

### Extracts contact score with Categorical Jacobian

Once the RCLM is trained, we can obtain a function $f(X)$ that maps an RNA sequence of length $L$ into a position-wise distribution matrix with a shape of $L \times 4$. Based on that, we construct a coupling tensor $J$ with a shape of $L \times 4 \times L \times 4$, defined as:

$$J_{i,a,j,b} = [f(X(x_i \to N_a))]_{j,b} - [f(X)]_{j,b}$$

(3)

Here, $x_i \to N_a$ means performing a mutation at $i$-th position into $a$-th nucleotide type $N_a$.

We then transform the coupling matrix $J$ into an intermediate $L \times L$ contact matrix $M$ by summing the squared values over the nucleotide dimension:

$$M_{i,j} = \left( \sum_{a=1}^{4} \sum_{b=1}^{4} (J_{i,a,j,b})^2 \right)^{\frac{1}{2}}$$

(4)

Finally, the Average Product Correction is then applied to remove the background noise and to obtain the final contact map.

$$C_{i,j} = M_{i,j} - \frac{\sum_{i'}^{L} M_{i',j} \sum_{j'}^{L} M_{i,j'}}{\sum_{i'}^{L} \sum_{j'}^{L} M_{i',j'}}$$

(5)

### RNA structure representation learning with RNA Transformer Block

After training RCLM, the sequence and pair representations obtained from its final layer are utilized as the input for RNA structure prediction. The backbone framework is the complete RNA Transformer Block, as illustrated in Fig 1C. The input sequence representation is first processed through an Attention module, where a bias term, derived from the linear projection of the input pair representation, is added to the attention weights. This bias can effectively incorporate pairwise information to the sequence modeling. The resulting sequence representation is then passed through a Transition Module to generate the final output sequence representation.

A separate copy of the output sequence representation services as the input of a set of pair processing modules, including Triangle Multiplication Outgoing, Triangle Multiplication Incoming, Triangle Attention StartingNode, Triangle

Attention EndingNode, and a final Pair Transition Module. In our implementation, the dimensions of sequence and pair representations are uniformly set to 64. For sequence representation learning, we set the number of attention heads to 8, with each head having a channel size of 8. Due to the high memory requirements of pair-related modules, however, we reduce the number of heads to 4. After passing through 16 RNA Transformer Blocks, we obtain the sequence and pair representations, $s \in R^{L \times 64}$ and $z \in R^{L \times L \times 64}$, which will be used for subsequent structure prediction.

During sequence-to-structure representation learning in DRfold2 through RNA Transformer Blocks, we introduce an additional attention dropout in regular attention operations. We expect that such a technique will prevent the model from over-relying on specific input tokens in the context. Specifically, when updating the token representations, we randomly set 25% of the attention weights to zero in the attention map. To maintain stability, we normalize the masked attention map by scaling the remaining attention weights based on their sum along the weighting dimension. During inference, we disable the attention dropout operation, allowing the model to fully utilize the input context.

**End-to-end RNA structure prediction with the Denoising Structure Module**

The sequence and pair representations obtained from the last RNA Transformer Block will be utilized as the inputs of the DRSM, as illustrated in Fig 1D. Each input representation first passes through an independent LayerNorm layer. An Invariant Point Attention (IPA) Module [20] iteratively takes the normalized features as inputs to get nucleotide-wise features, which will be subsequently used to predict nucleotide-wise rotation matrices and translation vectors.

Another required input for the IPA Module is the initial structural conformation. We apply black-hole initialization [20], where each nucleotide's initial state is defined by an identity rotation matrix $R_0 = I$ and a zero-translation vector $t_0 = 0$. During training, we employ a denoising strategy by adding controlled perturbations to these initial transformations. For rotation matrices, we apply weighted spherical interpolation $R_{noised} = slerp(I, R_{random}, w)$, where $R_{random}$ is a random rotation matrix, and $w = 0.1$ is the interpolation weight. For translation vectors, we add random noise and $t_{noised} = t_0 + \varepsilon$, where $\varepsilon \sim N(0, 0.1)$ denotes noise drawn from a normal distribution with mean 0 and scale 0.1. Given the high flexibility of RNA molecules, such controlled noise injection is expected to help improve model robustness during training and prediction. The parameters $w$ and $\varepsilon$ were chosen conservatively to introduce minor perturbations to the input without destroying the structural signal. The module is inspired by the denoising autoencoder [54] principle, i.e., DRSM will learn to map a distribution of perturbed initial conformations to the correct structure, rather than relying on a single fixed initialization as in AlphaFold2. This noise-conditioned training expands the diversity of the input space and consequently improves generalization. Such improvement is supported by the ablation (no-noise baseline), which shows a minor decrease in TM-score (S11 Fig). A related idea is also observed in AlphaFold3 [31] in the "ref_pos" feature, in which the reference conformer is randomly rotated and translated during training. This further supports the effectiveness of introducing stochastic perturbations into the initial structural states.

Besides the initial rotation matrices and translation vector, the local nucleotide conformation ($\vec{r}$) of three atoms, i.e., P, C4', and glycosidic N atoms of the nucleobase, should also be pre-determined in the structure module. Following our previous approach [21], symmetric orthogonalization [55] is utilized to obtain the coordinates for each atom from the ideal A-form RNA helix structure. Such orthogonalization has proven to have only half of the reconstruction error compared to the Gram-Schmidt process used in AlphaFold models [56].

In addition to the End-to-End structure models, DRfold2 also generates a set of inter-nucleotide distance predictions based on the pair representations from the last RNA Transformer Block, through a linear layer. Accordingly, the final loss function of DRfold2 for structure prediction consists of two terms:

$$Loss_{tot} = Loss_{structure} + 0.2 Loss_{dist} \tag{6}$$

where

$$\text{Loss}_{\text{structure}} = \text{gFAPE}\left(T, T^{\text{exp}}, \lambda, d^{\text{cut}}\right) - 0.1 * \text{gFAPE}\left(T, T_0, 1.0, 2.0\right) \tag{7}$$

The first term in Eq (4), i.e., generalized FAPE loss (gFAPE), quantifies the deviation between the predicted global nucleotide-wise rigid transformation ($T$) and the experimental global nucleotide-wise rigid transformation ($T^{\text{exp}}$), which is defined as,

$$\text{gFAPE}\left(T, T^{\text{exp}}, \lambda, d^{\text{cut}}\right) =$$
$$\sum_i \sum_j \lambda_{ij} \min\left(d_{ij}^{\text{cut}}, \sqrt{\left\| T_i^{-1}\left(T_j\left(\vec{r}\right)\right) - T_i^{\text{exp}-1}\left(T_j^{\text{exp}}\left(\vec{r}\right)\right) \right\|^2 + \epsilon}\right) \tag{8}$$

where $\lambda_{ij}$ represents the inter-nucleotide re-weighting factor and $d_{ij}^{\text{cut}}$ denotes the deviation cut-off parameter between nucleotides $i$ and $j$. If the inter-nucleotide distance in the experimental structure is below 20 Å, we set $\lambda_{ij} = 3.0$ and $d_{ij}^{\text{cut}} = \infty$. Otherwise, they are set to 1.0 and 30 Å, respectively. The value of $\epsilon$ is fixed at 1e−3. The subtraction of the second term in Eq (4) encourages deviation from the initial black-hole transformation ($T_0$). Our experiments indicate that incorporating both the re-weighting scheme and the subtraction term accelerates the training process.

The second loss in Eq (3) for distance terms is defined as

$$\text{Loss}_{\text{dist}} = \sum_{i,j} \sum_{g \in G} \log\left(p_{ij}^g\right) \tag{9}$$

where $G$ includes the set of pairwise distances between P, C4', and N atoms, respectively, for positions $i$ and $j$. The distance values are evenly divided into 38 bins spanning 2–40 Å, with two additional bins for distances <2 and >40 Å. $p_{ij}^g$ denotes the predicted probability in the respective distance bin $g$.

We trained four randomly initialized models for approximately 500,000 steps using a crop size of 256 tokens. The models were then trained for an additional 100,000 steps with a 384-token crop size. For each model, we retained the last 20 checkpoints saved at ~2,000-step intervals. This resulted in 80 checkpoints in total (4 models × 20 checkpoints each), which were used to create a conformational pool for further optimization.

## Structure optimization through CSOR protocol

During inference, for each input sequence, a collection of $N = 80$ end-to-end structures and distance probability maps is generated, where the final predicted models are obtained through the CSOR protocol (Clustering, Scoring, Optimization and Refinement), as illustrated in Fig 1E.

For **scoring**, we construct a global energy function $E_{\text{global}}$ which consists of two deep learning-based energy terms:

$$E_{\text{global}} = E_{e2e} + E_{\text{dist}} \tag{10}$$

The first energy term incorporates the conformational information through gFAPE:

$$E_{e2e} = \sum_{n=1}^{N} \text{gFAPE}\left(T^{\text{conf}}, T^n, w^n, inf\right) \tag{11}$$

where $T^{conf}$ represents the transformations of the given conformation to be scored/optimized; $T^n$ is the predicted conformation of $n$-th out of 80 predicted structures; and $w^n$ is the predicted pairwise error estimation score of $n$-th checkpoint. The second term in Eq (7) is defined by

$$E_{dist} = \sum_{n=1}^{N} \sum_{ij} \sum_{g \in G} \log \left( \frac{P_{ij}^{ng}\left(d_{ij}^{g}\right) + 0.001}{P_{ij}^{ng_M} + 0.001} \right)$$

(12)

where $P_{ij}^{ng}\left(d_{ij}^{g}\right)$ is the predicted probability of $n$-th prediction, given distance $d_{ij}^{g}$ for the $g$-distance bin. $P_{ij}^{ng_M}$ is the corresponding probability of the last distance bin ($g_M$) below the upper threshold (40 Å).

Based on the lowest $E_{global}$, we select the 5 predicted structures and their distance maps. A L-BFGS based **optimization** process [57] is performed under the guidance of $E_{opt} = E_{global}$, but with $N$ converted from 80 to 5 selected structures. Note that for targets larger than 150 nucleotides, we filter out raw conformations whose number of base pairs is lower than 90% of the highest one in our pool. We find it helpful for stabilizing the optimization. We observed that the top 10% of conformations ranked by base-pair count generally have higher average TM-scores (S12 Fig). This improvement is more pronounced for targets longer than 150 nt, with a ΔTM-score of 0.026, compared to 0.012 for shorter targets. We hypothesize that filtering by base-pair count preferentially selects conformations that better capture regularized structural motifs, thereby stabilizing the optimization. The final conformation with the lowest $E_{opt}$ will be considered as the *first* predicted model.

In order to find possible alternative/variant conformations from our prediction pool, we performed a basic **clustering** step by SPICKER algorithm [58]. Each cluster center will also go through the same scoring and optimization procedures described above. Eventually, up to 5 models will be obtained, and each of them will be **refined** by RNArefine. Here, RNArefine (available at https://zhanggroup.org/RNArefine/) is a new protocol which first infers base-pairing and base-stacking interactions from the initial structure. Built on these interactions and a physics-based force field, it then performs atomic-level refinement through a hybrid Monte Carlo sampling procedure followed by L-BFGS energy minimization, iteratively improving both local geometry and global structural consistency.

## Supporting information

**S1 Table. TM-score and RMSD of the first model by DRfold2 and AlphaFold3 for all 41 test targets.** Bold fonts highlight better performance in each subcategory.
(XLSX)

**S2 Table. Detailed comparison of AlphaFold3 and DRfold2 for different categories grouped by function and structure annotations.**
(XLSX)

**S3 Table. Physical geometry indexes of the first models by DRfold2 and experimental structurers.**
(XLSX)

**S4 Table. Average and Z-score based relative group performance of first model for TM-score for 20 RNA monomer targets (<400 Nts) in CASP16.** Only the top 20 groups are listed. Z-scores are calculated using a minimum Z-score cutoff >0.
(XLSX)

**S5 Table. Detailed technical comparison between DRfold and DRfold2 across multiple categories.**
(XLSX)

**S1 Fig. Box plots comparing of full-atom RMSD of control methods and DRfold2 respectively.** Green points indicate means and white horizontal lines show medians. Underlying numerical data for this figure can be found in S1 Data (see sheets "S1_Data_S1").
(TIF)

**S2 Fig. TM-score and RMSD comparisons between DRfold2 and AlphaFold3 on the novel-fold RNA targets. (A)** TM-score comparison. **(B)** RMSD comparison. Underlying numerical data for this figure can be found in S1 Data (see sheets "S1_Data_S2").
(TIF)

**S3 Fig. Structure visualization of the predicted conformations of target 8vci_A generated by DRfold2 before and after refinement. (A)** Prediction without refinement. **(B)** Prediction after refinement. Red dots indicate D&L interlace points. The structural visualizations are screenshots obtained from the RNAspider web server.
(TIF)

**S4 Fig. Box plots comparing of MCQ of control methods and DRfold2 respectively.** Green points indicate means and white horizontal lines show medians. Underlying numerical data for this figure can be found in S1 Data (see sheets "S1_Data_S4").
(TIF)

**S5 Fig. Box plots comparing of sequence recovery rate of RNA-FM, RiNALMo, RNAErnie, RCLM, and RCLM-M, respectively.** Green points indicate means and white horizontal lines show medians. Underlying numerical data for this figure can be found in S1 Data (see sheets "S1_Data_S5").
(TIF)

**S6 Fig. Examples of attention-based nucleotide-masking analysis for canonical and non-canonical RNA interactions. (A)** Results for 8HZD chain A. **(B)** Results for 8Z8U chain A. Nucleotides in red indicate the masked positions. Green nucleotide boxes represent attention magnitudes, where deeper green corresponds to higher attention weights. Blue arcs denote canonical base pairs, red arcs denote non-canonical pairs, and magenta arcs highlight interactions involving the masked nucleotide.
(TIF)

**S7 Fig. Recovery rates of canonical and non-canonical interactions by RCLM.** Underlying numerical data for this figure can be found in S1 Data (see sheets "S1_Data_S7").
(TIF)

**S8 Fig. Average TM-score of the first predicted models on 20 RNA monomer targets (<400 Nts) by CASP16 groups versus the dissimilarity to AlphaFold3.** For each group, dissimilarity = 1-TM_M where TM_M is the maximum TM-score between the first model and the five models from AlphaFold3.
(TIF)

**S9 Fig. Running time analysis of deep learning inference and post-processing steps across RNA lengths.** Underlying numerical data for this figure can be found in S1 Data (see sheets "S1_Data_S9").
(TIF)

**S10 Fig. TM-score comparison between DRfold and DRfold2 on the 41 test RNA structures.** Underlying numerical data for this figure can be found in S1 Data (see sheets "S1_Data_S10").
(TIF)

**S11 Fig. TM-score and RMSD comparisons between models trained with and without the denoising strategy. (A)** TM-score comparison (0.324 with denoising versus 0.316 without). **(B)** RMSD comparison (14.827 with denoising versus 14.927 without). Green points denote the mean values, and white horizontal lines mark the medians. Underlying numerical data for this figure can be found in S1 Data (see sheets "S1_Data_S11").
(TIF)

**S12 Fig. TM-score improvements by filtering based on base pair counts across targets for different lengths.** Underlying numerical data for this figure can be found in S1 Data (see sheets "S1_Data_S12").
(TIF)

**S1 Data. Underlying numerical data for Figs 2–6 and S1, S2, S4, S5, S7, S9–S12.**
(XLSX)

## Acknowledgments

We thank Drs. Robin Pearce and Chengxin Zhang for discussions.

## Author contributions

**Conceptualization:** Yang Zhang.

**Data curation:** Yang Li, Chenjie Feng.

**Investigation:** Yang Zhang.

**Methodology:** Yang Li.

**Resources:** Xi Zhang.

**Software:** Yang Li, Xi Zhang, Sho Tsukiyama.

**Supervision:** Yang Zhang.

**Validation:** Yang Li, Chenjie Feng, Duanyu Feng.

**Visualization:** Yang Li, Duanyu Feng.

**Writing – original draft:** Yang Li.

**Writing – review & editing:** Chenjie Feng, Xi Zhang, Sho Tsukiyama, Duanyu Feng, Yang Zhang.

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
