## [Editor Report · Decision Letter 0]

21 Jul 2025

Dear Dr Li,

Thank you for submitting your manuscript entitled "Composite likelihood language model enables efficient and accurate single-sequence RNA structure prediction" for consideration as a Methods and Resources article by PLOS Biology. Please accept my apologies for the delay in getting back to you as we consulted with an academic editor about your submission.

Your manuscript has now been evaluated by the PLOS Biology editorial staff, as well as by an academic editor with relevant expertise, and I am writing to let you know that we would like to send your submission out for external peer review.

Once your full submission is complete, your paper will undergo a series of checks in preparation for peer review. After your manuscript has passed the checks it will be sent out for review. To provide the metadata for your submission, please Login to Editorial Manager (https://www.editorialmanager.com/pbiology) within two working days, i.e. by Jul 23 2025 11:59PM.

Kind regards,

Richard

Richard Hodge, PhD

rhodge@plos.org

PLOS

---

## [Decision Letter · Decision Letter 1]

28 Aug 2025

Dear Dr Li,

Thank you for your patience while your manuscript "Composite likelihood language model enables efficient and accurate single-sequence RNA structure prediction " was peer-reviewed at PLOS Biology. Please accept my sincere apologies for the delays that you have experienced during the peer review process. Your manuscript has now been evaluated by the PLOS Biology editors, an Academic Editor with relevant expertise, and by three independent reviewers.

In light of the reviews, which you will find at the end of this email, we would like to invite you to revise the work to thoroughly address the reviewers' reports.

As you can see, the reviewers are generally positive about your improved method but raise some overlapping concerns. Specifically, they note that AlphaFold3 is excluded from the primary comparative figures and Reviewer #2 raises concerns that the central claim of being a "single-sequence" method is not fully leveraged or validated against MSA-dependent methods. In addition, the reviewers raise concerns about the contribution and reporting of the RCLM component to the model.

Given the extent of revision needed, we cannot make a decision about publication until we have seen the revised manuscript and your response to the reviewers' comments. Your revised manuscript is likely to be sent for further evaluation by all or a subset of the reviewers.

**IMPORTANT - SUBMITTING YOUR REVISION**

*Re-submission Checklist*

*Published Peer Review*

*PLOS Data Policy*

*Blot and Gel Data Policy*

Best regards,

Richard

Richard Hodge, PhD

rhodge@plos.org

REVIEWS:

Reviewer #1: Li et al. introduce DRfold2, a deep learning model for ab initio prediction of 3D structures of single-stranded RNAs, building on its predecessor, DRfold. A key innovation of DRfold2 is the integration of the RNA Composite Language Model (RCLM), which captures rich, sequence-dependent contextual features and significantly enhances structural modeling performance. DRfold2 demonstrates high accuracy compared to existing methods across diverse benchmarks, including RNA-Puzzles and CASP, with notable improvements in modeling complex and large RNA molecules. The paper is clear and extremely well illustrated. However, I have several questions and comments to the authors, that I would like them to address.

(1) Recently, several studies have emerged in which RNA-Puzzles/CASP assessors evaluated the quality of RNA structure predictions not only based on their similarity to reference structures but also considering the geometry and topology of the predicted structural models. For example, in the study by Carrascoza et al., 2022, titled "Evaluation of the stereochemical quality of predicted RNA 3D models in the RNA-Puzzles submissions," the authors pointed out that many models exhibit incorrect geometry (bond lengths, planarity, bond angles). Meanwhile, Gren et al., 2024, in their work "Knotted artifacts in predicted 3D RNA structures," examined the predicted models for topological irregularities (entanglements of structural elements and topological knots). I believe that post-prediction analysis of geometry and topology should become a standard and good practice among predictors and authors of new prediction tools. Therefore, I strongly suggest performing such analyses using available tools, such as MolProbity (for geometry) and RNAspider (for topology).

(2) An interesting metric used in RNA-Puzzles (see publications related to successive rounds of RNA-Puzzles) is the angular MCQ measure, which does not require structural alignment and reflects similarity in torsion angle space. Torsion angles are widely used in the analysis and determination of structures based on NMR data; also an RNA language model has been developed that is based specifically on torsion angles. I suggest the authors added a torsion angle-based metric to the set of evaluation measures they use for assessing predictions. Examples of easy-to-use implementations of such metrics can be found, among others, in the RNAtango web service.

(3) The TM-score values for the obtained predictions are poor. In the protein field, it is generally accepted that a TM-score < 0.5 indicates negligible similarity between the model and the target and is equivalent to a random prediction. TM-score values around 0.3 are therefore not evidence of correct or accurate predictions - quite the opposite - they indicate that the predictions are inaccurate. How do the authors comment on this fact?

(4) The evaluation of the predicted models is very coarse-grained if it reduces the entire nucleotide to a single representative atom. The same value of a given metric is interpreted very differently when calculated for a full-atom structure compared to a coarse-grained one. For example, a value indicating a rather poor-quality coarse-grained model may actually be quite good for a full-atom model, especially if the metric is distance-based and the evaluation involves large molecules. Applying the same scoring function to both full-atom and coarse-grained models yields completely different metric values. These values are not directly comparable. Why don't the authors assess the similarity using full-atom structures?

(5) Typos and minor issues:

- "likelihood[26]" (no space between)

- "assumption, First" -> "assumption, first"

Reviewer #2: This paper introduces DRfold2, a deep learning framework for the single-sequence RNA 3D structure prediction. They introduce two things: first, a pre-trained RNA Composite Language Model (RCLM) that utilizes a composite likelihood objective to capture pairwise nucleotide interactions from sequence data alone more effectively; and second, a Denoising RNA Structure Module (DRSM) integrated into the pipeline. The authors conduct extensive benchmarks against several state-of-the-art methods on a non-redundant test set.

Despite the claimed strengths, the paper has several weaknesses that should be addressed.

1. AlphaFold3 is one of the current state-of-the-art methods, and its exclusion from the main comparative figures (e.g., Figure 2) makes it difficult for the reader to immediately contextualize DRfold2's performance. While the dedicated analysis in Figure 6 is valuable, integrating AlphaFold3 into the primary benchmark plots would provide more context for DRfold2 against its top competitor.

2. The paper's central claim of being a "single-sequence" method is not fully leveraged or validated against MSA-based approaches in their domain of weakness. The key advantage of a single-sequence model should be its robust performance on targets with few or no homologs. The study would be much more compelling if it included an analysis comparing DRfold2 against MSA-dependent methods (like trRosettaRNA or an MSA-enabled AlphaFold3) on targets binned by their MSA depth. Such an experiment would directly test and highlight the practical utility of the single-sequence approach in the very scenarios where it is most needed.

3. The contribution and rationale of the Denoising RNA Structure Module (DRSM) is not sufficiently justified. The paper lacks an ablation study to quantify the performance gain specifically attributable to this module. For instance, what is the accuracy of the model without the noise injection during training? Furthermore, the term "denoising" is misleading, as the method adds noise to the initial state (identity rotations and zero-translation vectors) and deterministically predicts the structure, rather than the common approach to iteratively remove noise from corrupted ground-truth structures. A more detailed explanation of why this noise-conditioned training is advantageous over simply learning the transformation from a zero-initialized state, as in AlphaFold2, is needed to fully motivate its design.

4. There is a notable omission of any quantitative data on computational efficiency. The title claims the method is "efficient," but without runtime information, particularly for involving the multi-decoy CSOR post-processing protocol. It is difficult for the research community to assess the method's practical inference time.

5. The RCLM part does not provide any details on the configurations of RCLM like number of parameters or layer number while such details exist for other methods. For RCLM architecture, it is confusing that Fig1.C denotes a 'Outer Product Mean' process, while it should be something used for MSA. For RCLM training, it is mentioned that supervising second-order interactions can better approximate full RNA likelihood, but (x_i, x_j) and (x_j, x_i) are the same pair, meaning that the matrix-based likelihood prediction is supervised by a symmetric matrix. What is the point of incoming/outgoing triangle update operation here then?

6. The difference between DRfold2 and DRfold should be clearly stated. The whole architecture looks similar despite the RCLM.

Reviewer #3: The manuscript introduces DRfold2, a deep learning framework for ab initio RNA structure prediction. Key innovations include: (1) A novel RNA Composite Language Model (RCLM) trained using composite likelihood maximization to capture co-evolutionary patterns from unannotated sequences. (2) A Denoising RNA Structure Module (DRSM) for end-to-end 3D structure prediction. (3)A CSOR protocol for model selection/optimization. DRfold2 claims superiority over state-of-the-art methods (e.g., trRosettaRNA, RhoFold, AlphaFold3) in global topology (TM-score) and secondary structure (INF metrics) across multiple benchmarks. It also highlights complementarity with AlphaFold3, showing improved accuracy when integrated. However, the intellectual contribution to understanding RNA 3D structure folding is not clear. And the training process may suffer from data leakage. Major and minor comments are listed as below:

Major Concerns:

1. The paper positions DRfold2 as a technical advance but fails to derive new RNA folding principles from the model. For example: Does RCLM reveal specific nucleotide interaction rules (e.g., tertiary motifs, ion-binding preferences)? How do denoising or composite likelihood strategies translate to biophysical knowledge? I recommend including analysis of RCLM attention weights or feature importance to extract interpretable biological rules.

2. The model training process is likely to suffer from data leakage. Both the training set and the test set are redundant. E.g., 2 CPEB3 ribozymes are in the test set; multiple tRNAs with very similar sequences are found in the training set. In particular, PreQ1 Riboswitch, CoV-SL5, Sars-Cov-2 FSE exist in both training set and test set. Therefore, the model training is not reliable and all these benchmark results cannot be trusted.

Technically speaking, the test set includes PDB entries after 2024 (e.g., 9KPO, released March 2025), but RCLM was trained on RNAcentral (release 22, circa 2022-2023). However, no validation that post-2024 structures (or their close homologs) were absent from training. Sequence identity cutoff (80%) is insufficient; structural homology could persist below this threshold.

3. The results provide inadequate statistical support. E.g., need statistical tests and p-values for P4L158, P4L164, P6L248, P7L296, P8L321, P8L329, P8L354, P9L391, P10L423, P10L433, P10L449, P22L825.

4. There should be clearer comparison with AlphaFold3 in Figures 2 and 3. There is no analysis of specific RNA categories (e.g., pseudoknots, riboswitches) where DRfold2 excels/fails vs. AlphaFold3. And the complementarity claims (Fig. 6) lack mechanistic explanation.

5. The contribution of RCLM to non-canonical interactions is unclear. RCLM improves Watson-Crick pairs (INF_wc=0.838) but performs poorly on non-canonical pairs (INF_nwc=0.121, Fig. 3). The manuscript should quantify RCLM's role in non-canonical accuracy. Why it can improve? The authors should also discuss if composite likelihood specifically aids non-Watson-Crick modeling.

Minor Concerns:

The authors can still improve technical clarity.

P21: The noise-injection protocol is vague. Specify how w=0.1 and ϵ∼N(0,0.1) were chosen.

P22: CSOR protocol: Clarify why base-pair count filtering (>150 nt) stabilizes optimization.

P15: Categorical Jacobian: Briefly define how it extracts contacts.

Typos and Inconsistencies:

Abstract: "denoise-based" should be "denoising-based".

P4L143: "despite the low degree of freedom" → specify degrees of freedom.

P18L746: "Rinalmo" → "RiNALMo" (throughout).

P8L324: "SCOR" → "CSOR" (inconsistency with Fig. 1E).

Ref 22 (Li et al., 2022): avoid in the text.

---

## [Decision Letter · Decision Letter 2]

12 Jan 2026

Dear Dr Li,

Thank you for your patience while we considered your revised manuscript "Composite likelihood language model enables efficient and accurate single-sequence RNA structure prediction" for publication as a Methods and Resources Article at PLOS Biology. Please accept my sincere apologies for the delays that you have experienced during this round of the peer review process. This revised version of your manuscript has been evaluated by the PLOS Biology editors, the Academic Editor and the original reviewers.

Based on the reviews, I am pleased to say that we are likely to accept this manuscript for publication, provided you satisfactorily address the remaining points raised by Reviewer #3. In addition, please also make sure to address the following data and other policy-related requests that I have provided below (A-G):

(A) We routinely suggest changes to titles to ensure maximum accessibility for a broad, non-specialist readership. In this case, we would suggest a minor edit to the title, as follows. Please ensure you change both the manuscript file and the online submission system, as they need to match for final acceptance:

“DRfold2 is a deep learning-based tool that enables efficient and accurate RNA structure prediction”

(B) You may be aware of the PLOS Data Policy, which requires that all data be made available without restriction: http://journals.plos.org/plosbiology/s/data-availability. For more information, please also see this editorial: http://dx.doi.org/10.1371/journal.pbio.1001797

-Supplementary files (e.g., excel). Please ensure that all data files are uploaded as 'Supporting Information' and are invariably referred to (in the manuscript, figure legends, and the Description field when uploading your files) using the following format verbatim: S1 Data, S2 Data, etc. Multiple panels of a single or even several figures can be included as multiple sheets in one excel file that is saved using exactly the following convention: S1_Data.xlsx (using an underscore).

-Deposition in a publicly available repository. Please also provide the accession code or a reviewer link so that we may view your data before publication.

Figure 2A-B, 3A-B, 3D-F, 4A-F, 5A-D, 5F-G, 6A-F, S1, S2, S4, S5, S7, S9, S10, S11, S12

(C) Please also ensure that each of the relevant figure legends in your manuscript include information on *WHERE THE UNDERLYING DATA CAN BE FOUND*, and ensure your supplemental data file/s has a legend.

(D) Per journal policy, if you have generated any custom code during the course of this investigation, please make it available without restrictions. Please ensure that the code is sufficiently well documented and reusable, and that your Data Statement in the Editorial Manager submission system accurately describes where your code can be found. More information on our Code Policy, what and how to share can be found here: https://journals.plos.org/plosbiology/s/code-availability

(E) Please ensure that your Data Statement in the submission system accurately describes where your data can be found and is in final format, as it will be published as written there.

(F) Please ensure that you are using best practice for statistical reporting and data presentation. These are our guidelines https://journals.plos.org/plosbiology/s/best-practices-in-research-reporting#loc-statistical-reporting and a useful resource on data presentation https://journals.plos.org/plosbiology/article?id=10.1371/journal.pbio.1002128

- If you are reporting experiments where n ≤ 5, please plot each individual data point.

(G) Please note that per journal policy, the species studied should be clearly stated in the abstract of your manuscript.

We expect to receive your revised manuscript within three weeks.

*Published Peer Review History*

*Press*

Best regards,

Richard

Richard Hodge, PhD

rhodge@plos.org

Reviewer remarks:

Reviewer #1: I have not further comments.

Reviewer #2: The authors have resolved my previous concerns.

Reviewer #3: I appreciate the authors' careful and detailed responses to the previous round of comments, as well as the additional analyses that have been incorporated into the revised manuscript. The revision has substantially improved the clarity of the work. Some conceptual and methodological points would benefit from further clarification to ensure that the conclusions drawn are fully supported.

1. Line 246: Demonstrating performance on targets with normalized TM-score ≤ 0.45 relative to the training set strengthens the argument that DRfold2 does not rely solely on trivial memorization. Nevertheless, regarding the inclusion of post-2024 PDB structures, a brief clarification of whether homologous sequences or highly similar families might already exist in RNAcentral (release 22) would improve transparency, even if full exclusion cannot be guaranteed.

2. Line 284: For the hybrid DRAF approach, a brief clarification of whether AlphaFold3 is used purely as an external structural prior or implicitly transferring training knowledge would help avoid confusion about fairness and information flow between models.

---

## [Editor Report · Decision Letter 3]

2 Feb 2026

Dear Dr Li,

On behalf of my colleagues and the Academic Editor, Yunsun Nam, I am pleased to say that we can accept your manuscript for publication, provided you address any remaining formatting and reporting issues. These will be detailed in an email you should receive within 2-3 business days from our colleagues in the journal operations team; no action is required from you until then. Please note that we will not be able to formally accept your manuscript and schedule it for publication until you have completed any requested changes.

PRESS

Best wishes,

Richard

Richard Hodge, PhD

rhodge@plos.org

PLOS
